# *Brd4::Nutm1* fusion gene initiates NUT carcinoma in vivo

Dejin Zheng[1,2,*] , Ahmed A Elnegiry[1,2,8,*] , Chenxiang Luo[1,2,9,*], Mohammed Amine Bendahou[3], Liangqi Xie[3], Diana Bell[4], Yoko Takahashi[5], Ehab Hanna[5] , George I Mias[2,6] , Mayra F Tsoi[7] , Bin Gu[1,2]

**NUT carcinoma (NC) is an aggressive cancer with no effective treatment. About 70% of NUT carcinoma is associated with chromosome translocation events that lead to the formation of a *BRD4::NUTM1* fusion gene. Because the *BRD4::NUTM1* gene is unequivocally cytotoxic when ectopically expressed in cell lines, questions remain on whether the fusion gene can initiate NC. Here, we report the first genetically engineered mouse model for NUT carcinoma that recapitulates the human t(15;19) chromosome translocation in mice. We demonstrated that the mouse t(2; 17) syntenic chromosome translocation, forming the *Brd4::Nutm1* fusion gene, could induce aggressive carcinomas in mice. The tumors present histopathological and molecular features similar to human NC, with enrichment of undifferentiated cells. Similar to the reports of human NC incidence, *Brd4::Nutm1* can induce NC from a broad range of tissues with a strong phenotypical variability. The consistent induction of poorly differentiated carcinoma demonstrated a strong reprogramming activity of BRD4:: NUTM1. The new mouse model provided a critical preclinical model for NC that will lead to better understanding and therapy development for NC.**

## Introduction

NUT carcinomas (NCs), formerly named NUT midline carcinomas, are poorly differentiated squamous cell carcinomas (SCCs) that often arise within midline organs (1, 2, 3). Other primary sites, including skin (4), pancreas (5), pelvic organs (6, 7), and soft tissues (8, 9), have also been reported. They are aggressive tumors that typically present with metastatic disease at diagnosis (10). The prognosis of NC is dismal, with an overall survival time of 6.5 mo, even with intensive treatment, including surgery, radiation, and chemotherapy (10). NC is rare; a recent estimation suggests that NC accounts for ~0.21% of all cancers, with an estimated 3,500 new cases emerging in the USA each year (11). However, because NC lacks a distinctive diagnostic feature and is not widely recognized by many clinicians, the case rates are likely underestimated. NC is more prevalent in young people, causing a dramatic loss of life (12). Thus, it is critical to understand the pathogenesis of NC, which will lead to better treatment for NC patients.

Clinical studies have established a strong association of NC with chromosome translocation events that produce fusion genes involving a testis-specific gene known as *Nuclear protein in testis (NUTM1)* (13), with various epigenetic factors, including *BRD4* (70% of cases), *BRD3*, *NSD3*, and *ZNF532* (1, 4, 9, 14, 15, 16, 17). Limited genomic sequencing and karyotyping analysis suggested that these fusion genes, likely acting as an abnormal epigenetic modifier, can drive NC without other mutations in a quiet genomic landscape (18). At the molecular level, it was suggested that by recruiting the P300 acetyltransferase (HAT) to ectopic sites, BRD4::NUTM1 and the other fusion proteins drive malignant transformation by activating oncogenes and epithelial progenitor genes such as *MYC*, *SOX2*, and *TP63* (19, 20, 21, 22, 23).

However, when ectopically expressed in non-NC cell lines, *BRD4:: NUTM1* is unequivocally cytotoxic (2, 24, 25), contradictory to the proposed oncogenic functions. To reconcile these contradictory observations, we consider two possible explanations. First, many fusion oncogenes have been shown to require a specific cell lineage status to achieve cell transformation and are otherwise toxic to other cell types. This is coined as a concept of "cellular pliancy" first in pediatric cancer and then in cancer in general (25, 26, 27). Second, it could simply be that the overexpression system used on cell lines expresses the *BRD4::NUTM1* oncogene at a too high level. It is not uncommon to observe gene dosage–dependent toxicity of oncogenes (28, 29). Thus, the discrepancy in expression levels could be a simple explanation for the cytotoxicity of the *BRD4::NUTM1* oncogene.

To recapitulate the NC oncogenesis in an experimental model in which both critical parameters are controlled, we developed a

[1]Department of Obstetrics, Gynecology and Reproductive Biology, College of Human Medicine, Michigan State University, East Lansing, MI, USA  [2]Institute for Quantitative Health Science and Engineering, Michigan State University, East Lansing, MI, USA  [3]Infection Biology and Cancer Biology Program, Lerner Research Institute, Cleveland Clinic, Cleveland, OH, USA  [4]City of Hope Comprehensive Cancer Center, Pathology, Duarte, CA, USA  [5]Department of Head and Neck Surgery, the University of Texas MD Anderson Cancer Center, Houston, TX, USA  [6]Department of Biochemistry and Molecular Biology, College of Nature Science, Michigan State University, East Lansing, MI, USA  [7]Department of Pathobiology and Diagnostic Investigation, College of Veterinary Medicine, Michigan State University, East Lansing, MI, USA  [8]Home Institution: Department of Cytology and Histology, Faculty of Veterinary Medicine, Aswan University, Aswan, Egypt  [9]Home Institution: Center for Reproductive Medicine and Department of Gynecology & Obstetrics, the First Affiliated Hospital, Sun Yat-Sen University, Guangzhou, PR China

Correspondence: gubin1@msu.edu; tsoimayr@msu.edu
*Dejin Zheng, Ahmed A Elnegiry, and Chenxiang Luo contributed equally to this work

genetically engineered mouse model (GEMM). Using cre/lox-mediated chromosome translocation, the model ensured the recapitulation of human-mimicking chromosome translocation and established syntenic control of the resultant *Brd4::Nutm1* fusion gene. Taking advantage of the intrinsic low frequency of cre/lox-mediated inter-chromosome translocation (30, 31, 32, 33), the model recapitulated the stochastic nature of fusion gene formation. Using various cre drivers, we demonstrated a broad oncogenic activity of *Brd4::Nutm1* across tissues, covering all reported sites of human NC. *Brd4::Nutm1* drives aggressive SCC with various degrees of differentiation, consistent with human NCs. Furthermore, the model demonstrated a strong intrinsic metastasis potential of NC, recapitulating a key feature of human NC. Thus, we reported the first GEMM that recapitulates the syntenic chromosome translocation and oncogenic process of human NC. The model will create tremendous opportunities for understanding and developing treatments for this devastating disease.

## Results

### *Brd4::Nutm1* induces NC-like tumors from epithelial progenitor cells

We designed and developed a mouse model to model the NC initiation event using 2C-HR-CRISPR (34) (Fig 1A). Loxp sites were inserted into the appropriate introns of the mouse *Brd4* and *Nutm1* genes to allow stochastic induction of a reciprocal t(2;17) chromosome translocation forming the *Brd4::Nutm1* fusion gene by Cre recombinase. To track the development of NC in vivo using bioluminescence imaging (BLI), we inserted a Luc2TdTomato reporter cassette downstream of the endogenous *Nutm1* coding sequence, separated by a T2A self-cleaving sequence (35). As expected, the Luc2TdTomato reporter was expressed specifically in *Nutm1*-expressing post-meiotic spermatids (Fig S1A and B). For brevity, mice carrying both loxp sites and the reporter cassette linked to *Nutm1*$^{loxp}$ are designated NC translocators (NCT). From this point, unless otherwise explained, NCT$^{+/wt}$ refers to *Brd4*$^{loxp/wt}$; *Nutm1*$^{loxp/wt}$, whereas NCT$^{+/+}$ refers to *Brd4*$^{loxp/loxp}$; *Nutm1*$^{loxp/loxp}$.

Around 40% of human NCs are poorly differentiated SCC that arose in the head-and-neck region. They express markers reminiscent of basal progenitor cells (3, 36, 37, 38). Thus, we first tested NC induction using the *KRT14Cre* allele that expresses the Cre recombinase in ectoderm-derived basal progenitor cells (39) (Fig S1C). At weaning age, *KRT14Cre*$^{+/-}$; NCT$^{+/wt}$ mice developed BLI-positive head tumors at 100% penetrance (Figs 1B and S1D) (n > 50 mice), with some mice also developing skin tumors. These tumors grew exponentially (Fig 1C and D) and rapidly led to a moribund state (followed by humane euthanasia) (Fig 1E), consistent with the aggressive nature of human NC, likely because of the large tumor burden in the oral cavity causing obstruction of the upper airway and digestive tracts. Upon autopsy, tumors were identified in the oral cavity (mandible, maxilla, and tongue), salivary glands, eyelids, ear canal, and skin (Fig S1E); all of these locations have been reported in human patients and consistent with the expression pattern of *KRT14Cre*.

We validated the formation of the t(2;17) chromosome translocation by (1) duo-color FISH (DC-FISH) (representative image in Fig 1F, low-power image in Fig S2A); (2) Sanger sequencing of the junction sequence of DNAs extracted from three tumors (Fig S2B); and (3) karyotyping (Fig S2C). Furthermore, fusion calling from whole-genome sequencing of two paired datasets from tumor and normal liver DNA from the same tumor-bearing mice detected *Brd4::Nutm1* as the only recurrent fusion gene in tumor tissues (Fig S2D). At the gene expression level, fusion calling from RNA-sequencing data of six tumor samples again demonstrated *Brd4::Nutm1* fusion with the designed junction as the only recurrent fusion (Fig S2E and F). Furthermore, immunohistochemistry (IHC) analysis against NUTM1 detected the pervasive nuclear expression of the BRD4::NUTM1 protein in the tumors (Figs 1G, a and S2G). No BRD4::NUTM1 was expressed in NC surrounding normal epithelial or tumor-associated stromal cells (Fig 1G, b). In addition, Western blotting analysis confirmed the correct size of the BRD4::NUTM1 fusion protein in mouse NUT carcinomas (mNCs) (Fig S2H). We counted 3,500 normal oral mucosa cells on IHC samples from four NC-bearing mice and did not detect isolated NUTM1-positive cells (Fig S3), confirming the low frequency and stochastic nature of the chromosome translocation. The above-described data suggest that the t(2;17) chromosome translocation that creates *Brd4::Nutm1* can induce aggressive tumors from mouse epithelial basal cells. Because the mouse tumors satisfy two key diagnostic criteria applied to humans—the detection of the chromosome translocation by FISH and the detection of BRD4::NUTM1 proteins by IHC (15, 40)—we call these tumors mNC.

### mNCs present histopathological and molecular features of human NC

Most human NCs are categorized as poorly differentiated SCCs, characterized by primitive round cells with variable areas of abrupt squamous differentiation and keratinization (11). This heterogeneous phenotype has caused profound confusion in diagnosing NC. As shown in Figs 2A and B and S4A, a and b, mNCs developed in the head-and-neck locations present as SCCs with different degrees of differentiation. We did not observe a consistent correlation between the tumor location and the degrees of differentiation, which was consistent with human NCs. The tumors consist of large areas of basaloid (Fig S4A, c), polygonal (Fig S4A, d), or spindle-shaped (Fig S4A, e) cells. Many tumors exhibit areas of squamous cell differentiation and keratinization that occupy variable proportions of the tumor (Fig S4A, f). These features are consistent with the histopathological observations in the clinical setting. Furthermore, a board-certified veterinary pathologist scored 50 cases of mouse NC arising in the head-and-neck and skin sites and concluded that the mouse NCs present a wide range of degrees of squamous cell differentiation (Fig 2B), recapitulating the human pattern. Notably, even within a single mouse, individual tumors can display variable levels of keratinization and stromal infiltration (Fig 2C), demonstrating intra- and inter-tumor heterogeneity within the same genetic background. Thus, mNCs reproduced the morphological heterogeneity of human NC.

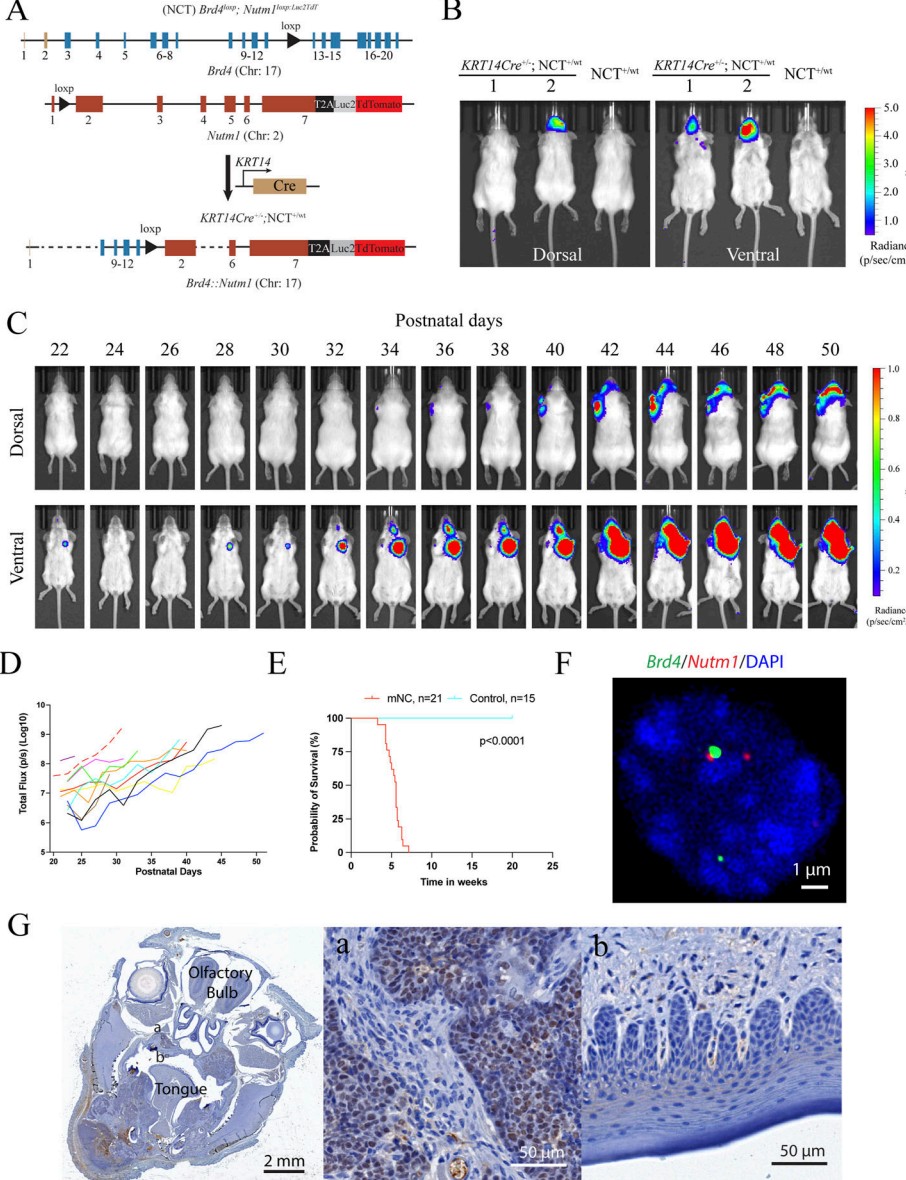

**Figure 1. Generation of the NC GEMM.**
**(A)** Genetic design of the NUT carcinoma translocator (NCT) and the generation of the *KRT14Cre*-induced mNC. **(B)** Representative BLI images of mice with head NC. **(C)** BLI images of the growth kinetics of mNCs. **(D)** Growth curves of mNCs in 11 mice quantified by BLI signals; each curve represents one mouse. **(E)** Kaplan–Meier survival curve of *KRT14Cre*-induced mNC mice. The *P*-value was calculated by the log-rank test. **(F)** DC-FISH of the *Brd4* (Green) and *Nutm1* (Red) on mNCs. **(G)** Representative NUTM1 IHC image of NC-bearing whole mouse head (leftmost panel), and zoom-ins on the tumor (a) showing pervasive NUTM1 staining, and normal oral mucosa (b) regions showing no NUTM1 staining.

A few key molecular features define human NC. First, the BRD4::NUTM1 proteins form nuclear foci, possibly as biological condensates mediated by the highly disordered sequences in NUTM1, in human NC (41, 42). We detected such foci in mNCs through both immunofluorescence (IF) and IHC in oral NC (Figs 2D and S4B). Second, human NC tissues and cell lines demonstrated nuclear foci of histone 3 lysine 27 acetylation (H3K27ac), likely caused by the recruitment of P300 Histone Acetyltransferase (HAT) by NUTM1 (19). We detected such foci in mNCs through IF (Figs 2E and S4C). Consistent with the proposed function that BRD4::NUTM1 promotes H3K27ac (19, 41), the fluorescent intensity of H3K27ac IF signals was significantly increased in NC cells (TdTomato positive) compared with surrounding non-NC cells (TdTomato negative) (Fig S4C and D). Third, human NCs express other hallmark genes, including the basal progenitor cell markers *TP63*, *KRT14*, *SOX2*, and

oncogene *MYC*. These genes have also been implicated as required for the progression of NC (20, 22, 23). Most of the oral mNC cells express P63, KRT14, and MYC, although the staining showed variable strength, indicating variable expression levels (Figs 2F and S4E). However, SOX2 is expressed in a small proportion of NC cells with a clustered pattern in all tested mouse NCs (Figs 2F and S4E). In clinical pathology practice, SOX2 is considered a stem cell marker in NC and is expressed in various proportions of NC cells (23, 43), consistent with the mouse data. Whether the SOX2-positive NC cells represent a cancer stem cell population is a critical question to be investigated in the future. Critically, all these markers stained positive in most of the basal cells within the normal oral mucosa, consistent with known patterns (Fig S4E). To understand whether the observed variation of the expression of key NC markers, including *Myc* and *Trp63*, and the variation in

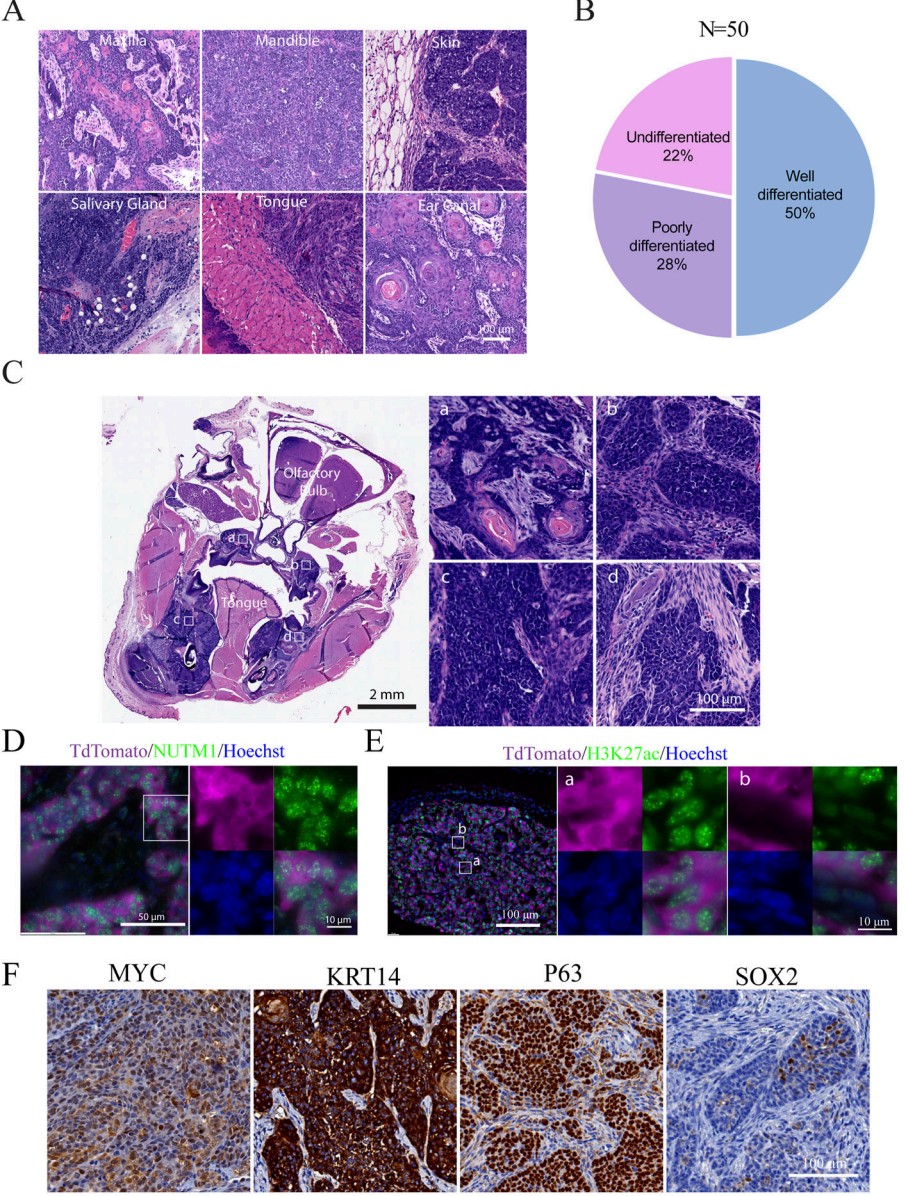

**Figure 2. Histological and molecular characteristics of mNCs.**
**(A)** Representative H&E staining images of *KRT14Cre* mNC tumors at different locations within the head-and-neck region. **(B)** Distribution of pathological features of mNCs. **(C)** Tumors formed in the same mouse presenting variable levels of differentiation. **(D)** Representative image of IF staining of NUTM1 in mNC showing nuclear foci. TdTomato staining demarcates tumor cells. **(E)** Representative images of IF analysis of H3K27ac in mNCs, depicting nuclear foci in tumor cells (a) versus a weaker diffused nuclear stain in peritumor stromal cells (b). **(F)** Representative IHC images of MYC, KRT14, P63, and SOX2 in oral mNCs.

the *Sox2*-expressing cell population are related to the expression level of the *Brd4::Nutm1* fusion gene, we analyzed their expression level using the bulk RNA-sequencing data (discussed in more details below). As shown in Fig S4F, although the expression levels of *Brd4::Nutm1* were consistent relative to normal oral mucosa tissues among six samples, *Myc* was only mildly up-regulated and showed strong variability between samples. *Trp63* was strongly up-regulated but also to a variable degree. *Sox2* was down-regulated, consistent with decreased proportions of *Sox2*-positive cells in mNCs compared with normal oral mucosa. No apparent correlation can be established between the expression levels of *Brd4::Nutm1* and any of the three markers, consistent with the reported phenotypical variability in human NC and implies a complex regulation of NC markers for *Brd4::Nutm1*, meriting future investigations.

## mNCs are aggressive poorly differentiated cancer at the transcriptomics level

To gain a global understanding of how transcriptomics reprogramming promoted aggressive mNC, we performed bulk RNA-sequencing analysis on six normal oral mucosa (control) and six oral mNC (tumor) samples. As shown in Fig S5A–D, the RNA-sequencing data are of high quality, with a clear separation of the six normal oral mucosa and the six mouse NC repeats. The differential expression analysis revealed 3,065 up-regulated genes and 2,654 down-regulated genes in mNCs, demonstrating pervasive transcriptomic reprogramming (Fig 3A). Gene set enrichment analysis (GSEA) against the Molecular Signatures Database (MSigDB) hallmark gene set collection revealed the activation of many pathways that may contribute to an aggressive phenotype, including cell proliferation

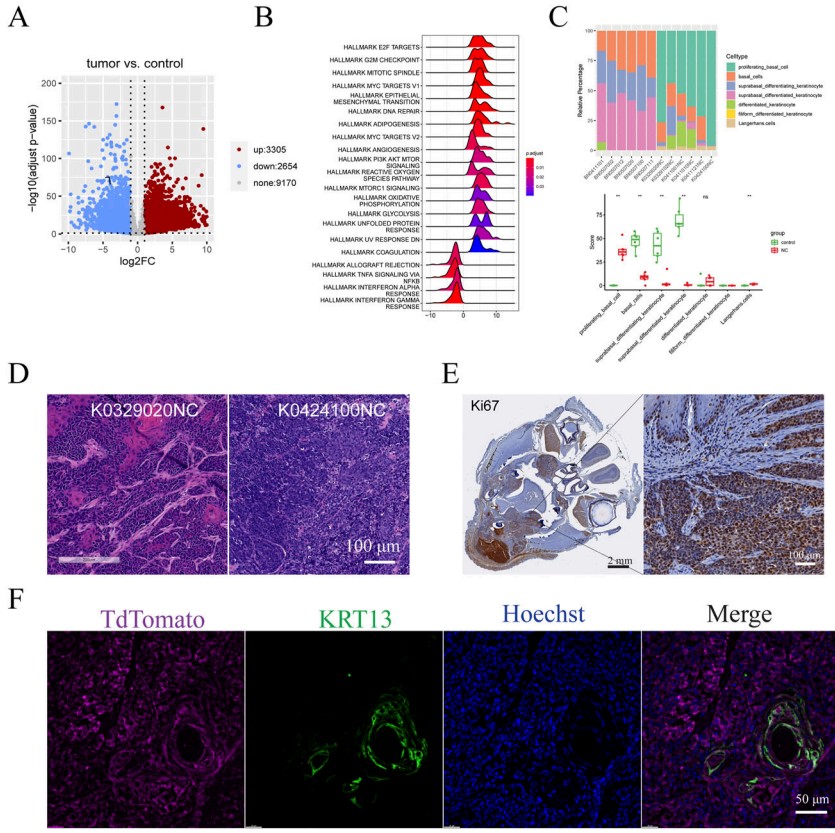

**Figure 3. Brd4::Nutm1 induces gene expression reprogramming to a poorly differentiated cellular status.**
**(A)** Volcano plot of differentially expressed genes in mNCs compared with normal oral mucosa. More than 3,000 genes are up-regulated (red), and 2,654 genes are down-regulated (blue) in NC mice compared with controls. Cutoff: abs(LFC) > 1 and adjusted $P < 0.05$. LFC, log2 fold change. **(B)** Ridgeline plot of 21 significantly enriched hallmark gene sets by GSEA on all significantly regulated genes in mNCs versus controls. Gene sets with adjusted $P < 0.05$ were considered significantly enriched. **(C)** CIBERSORT deconvolution of the transcriptomes of mNCs according to the Tabula Muris dataset. Upper: stacked barchart of relative proportions of different cell types in tumor and normal samples. Lower: boxplot of absolute scores of different cell types in tumor and normal groups. Data are presented as the median, and the 25th and 75th percentiles. $P$-values are calculated from Wilcoxon's rank sum tests. ns: $P > 0.05$; **: $P \leq 0.01$. **(D)** Representative histology images of mNCs that enrich the proliferative basal cell profile at different levels. **(E)** Representative Ki67 IHC image of mNCs. **(F)** Representative IF image of NC cells (tdTomato) and the ones expressing keratinization marker KRT13.

and mitosis, DNA repair, MYC target pathways, and metabolic reprogramming (Fig 3B). Notably, down-regulated genes are specifically enriched for immunostimulatory pathways (Fig 3B).

To understand whether the poorly differentiated morphological traits are represented at the transcriptomics level, we used CIBERSORT to deconvolute the bulk RNA-sequencing data using published single-cell RNA-sequencing (scRNA-seq) signatures (44, 45). As reference data, we used the published scRNA-seq data on the oral epithelial cells from the *Tabula Muris* project (46), which represents the cells from which the NC tumors are derived. As shown in Fig 3C, compared with the normal oral mucosa samples, there is a strong enrichment of the proliferating basal cell expression profile in the NC samples. CIBERSORT analysis using another independently produced scRNA-seq data of mouse oral mucosa as reference produced a similar result (47) (Fig S5E and F). The CIBERSORT results are consistent with the histological features of these tumors; for example, case K0329020NC showed more differentiated histology, whereas K0424100NC is more undifferentiated (Fig 3D). At the molecular level, the overwhelmingly enriched proliferating basal cell profile is consistent with the pervasive expression of basal cell markers (Fig 2). Consistent with its highly proliferative nature, mNC cells overwhelmingly expressed the cell proliferation marker Ki67 (Fig 3E). In addition, the differentiated keratinocyte marker KRT13 was only detected in small regions with apparent keratinization (Fig 3F). Thus, the *Brd4::Nutm1* expression trapped most NC cells in a proliferative progenitor status, likely contributing to its aggressive nature.

## Brd4::Nutm1 can induce aggressive malignancy from a broad range of tissues

In humans, although most NCs were detected within the head-and-neck or thoracic tissues (3), there were reports of carcinoma carrying the *BRD3/4::NUTM1* fusion genes in many other epithelial tissues, such as the pancreas (5), the kidney (48), and the pelvic organs (7); and even mesenchymal tissues such as undifferentiated soft tissue tumors (8). This broad tissue distribution is a unique property of NC compared with many other fusion gene-driven cancers and indicates that the *BRD4::NUTM1* fusion gene may transform a broad range of cell types covering all three germ layers. However, because of the limited number of cases, it is unclear whether the reported *NUTM1* fusion genes are the cancer driver or background mutations irrelevant to oncogenesis in these cases. We tested it using the GEMM.

There were two cases of NC involving the pancreas reported in very young children (5). The congenital nature of these cases suggests that an embryonic progenitor was transformed. Thus, we crossed the *Pdx1Cre* allele (49), which drives recombination in endoderm-derived embryonic ductal progenitor cells into the NCT mouse line. *Pdx1Cre*$^{+/-}$; NCT$^{+/wt}$ mice developed pancreatic tumors with 100% penetrance (18/18) (Figs 4A and S6A). The pancreatic mNCs are fast-growing SCCs presenting all histopathological and molecular markers as oral mNCs (Figs 4B and C and S6B). Beyond the epithelial tissue, a few cases of soft tissue tumors have been reported to carry *BRD3/4::NUTM1* fusion genes, suggesting that the

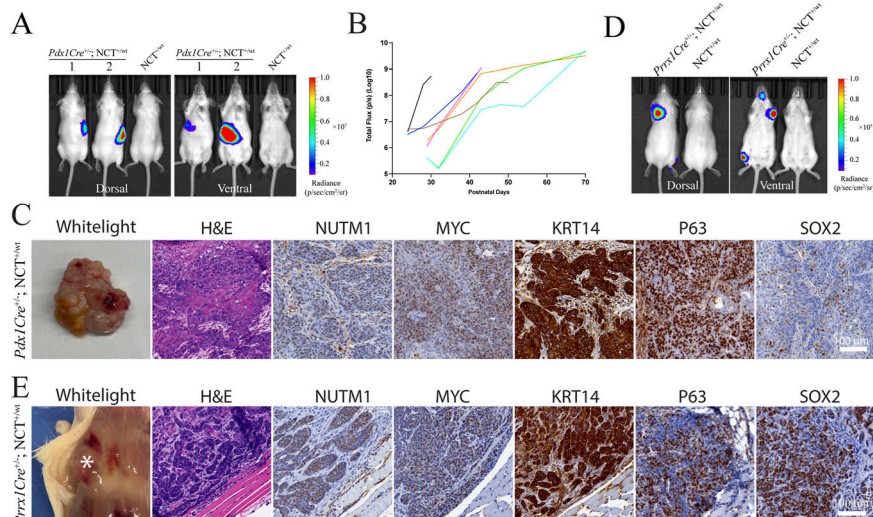

**Figure 4. *Brd4::Nutm1* initiates pancreatic and soft tissue tumors.**
**(A)** Representative BLI images of mice with pancreatic mNCs. **(B)** Growth curves of pancreatic mNCs in seven mice quantified by BLI signals; each curve represents one mouse. **(C)** Representative macroscopic, H&E, and IHC images of NUTM1, MYC, KRT14, P63, and SOX2 in pancreatic mNCs. **(D)** Representative BLI images of mice with soft tissue mNCs. **(E)** Representative macroscopic, H&E, and IHC images of NUTM1, MYC, KRT14, P63, and SOX2 in soft tissue mNCs.

fusion gene can also collaborate with mesoderm-derived mesenchymal gene regulatory networks (GRNs) to drive NC (8). To test this, we used a *Prrx1Cre* to drive the chromosome translocation in the fetal limb mesenchymal cells (50). These mice developed fast-growing tumors within limbs at 100% penetrance (9/9) (Figs 4D and S6C). The tumors are fast-growing and appeared morphologically as poorly differentiated SCC. They presented all histopathological and molecular markers of NC (Figs 4E and S6D). As the pancreatic ductal progenitors normally initiate adenocarcinoma, and mesenchymal progenitors normally initiate sarcomas, the ability to induce tumors morphologically consistent with poorly differentiated SCC-like tumor consistently indicated a strong reprogramming activity of *Brd4::Nutm1* (49, 51). However, because of the still limited number of cases and overall late-stage tumors analyzed in this study, we cannot rule out a progressive transformation to an SCC-like phenotype at a later stage of NC progression or the possibility of nonspecific expression *Pdx1Cre* or *Prrx1Cre* in rare squamous epithelial progenitors.

The clonal and stochastic nature of the translocation induction provided a unique advantage of modeling the reality of stochastic gaining of the fusion gene across tissues in humans. We crossed the NCT mouse line with the *NLSCre* mouse line that constitutively expresses Cre in all mouse tissues (52). This cross will induce *Brd4::Nutm1*, forming chromosome translocation stochastically across tissues. The $NLSCre^{+/-}$; $NCT^{+/wt}$ mice developed tumors detected by BLI signals at 100% penetrance (25/25) (Fig S7A). Autopsy guided by BLI imaging identified tumors in a broad range of anatomical locations, including the head-and-neck region (oral mucosa, paranasal sinuses, tongue, salivary glands), thoracic cavity, stomach, kidney, ovary, skin, and limb soft tissues (Figs 5A and S7A, and Table S1), covering all reported anatomical locations of human NC. Morphologically, most of these tumors presented as SCC with differing degrees of differentiation (Figs 5B and C and S7B and C), confirming a consistent phenotype. Notably, although the *NLSCre*-driven mNCs consistently express BRD4::NUTM1, MYC, and KRT14, the expression of P63 and SOX2 is variable (Fig 5B and C; staining patterns in correspondent normal tissues are shown in Fig S7D). The

inter-tumor heterogeneity of these progenitor transcription factors, with no clear correlation with anatomical locations, is consistent with pathological observations in human patients. Critically, atypical histological phenotypes were reported in isolated human NC cases, including glandular and chondrogenic metaplasia (53). Consistently, regions of chondrogenic and glandular transitions were observed in some mNCs (Figs 5D and S7E). The new GEMM provided a critical tool for investigating the reprogramming capability of *Brd4::Nutm1* to drive NC across tissues with remarkable heterogeneity.

**Mouse NCs have high metastasis potential**

Most human NC patients have metastatic disease at diagnosis, contributing significantly to its dismal prognosis (2, 54, 55). Of the more than 100 oral NC mice we analyzed, there were four cases of metastasis to proximal lymph nodes (Fig 6A), suggesting metastatic potentials of mouse NC. However, because the oral NCs lead to a quick death of mice, they may not allow the time for metastasis to establish and emerge. For further investigation, we focused on analyzing the *Pdx1Cre*-driven pancreatic NC, which survives longer. Pancreatic NC mice consistently develop broad dissemination and metastasis at 10 wk of age. Large numbers of dissemination nodules (carcinomatosis) develop across the mesentery membrane (Figs 6B and S8A). Distant micrometastasis was detected in anatomical sites, including the lung and liver (Fig 6B). The metastatic tumors express NC markers, including NUTM1, MYC, P63, KRT14, and contain populations of SOX2-positive cells (Fig 6C).

It is unclear why NCs are extremely metastatic. Desmoplasia, characterized by the recruitment of cancer-associated fibroblasts and pervasive deposition of extracellular matrix proteins, is strongly associated with metastatic potential in common cancers such as pancreatic adenocarcinoma (PDAC) (56, 57, 58). Desmoplastic histological features have been reported in a number of human NC case reports and are routinely observed in clinical diagnostic practice (59, 60, 61), yet they have never been characterized at the molecular level. Through RNA-sequencing analysis of

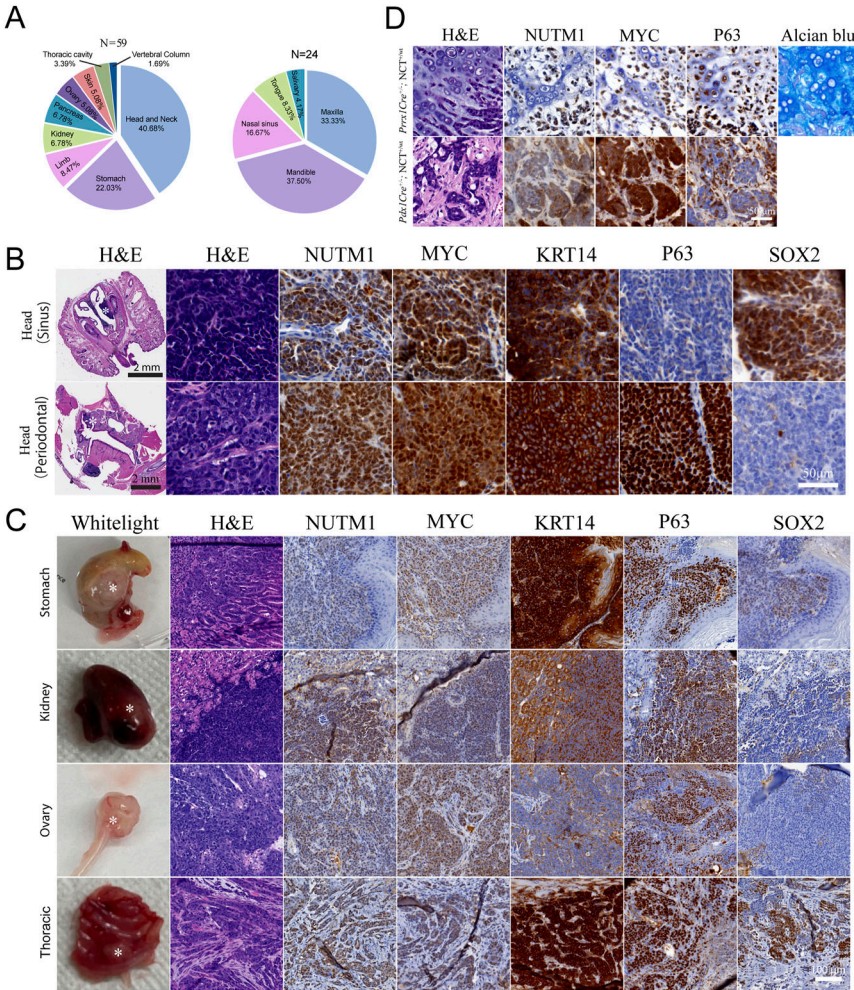

**Figure 5.** ***Brd4::Nutm1* initiates NC across tissues.**
**(A)** Tissue distribution of the *NLSCre*-driven mNCs; left: distribution throughout the body; right: distribution of the head-and-neck tumors. **(B)** Representative whole-head H&E, high-magnification H&E, and IHC images of nasal and oral *NLSCre*-driven mNCs. **(C)** Representative macroscopic, H&E, and IHC images of *NLSCre*-driven mNCs outside the head-and-neck region. **(D)** Representative H&E, IHC, and Alcian Blue cartilage staining images of chondrogenic and ductal metaplasia in mNCs.

mouse NC, we detected large increases in the expression of fibroblast genes such as *fibroblast activation protein (FAP)*, *smooth muscle alpha-actin (Acta2)*, and SM22alpha (*Tagln*) (Fig 7A). Mouse NC tissue also expresses high levels of ECM genes including *fibronectin (Fn1)*, *collagens* (17 types), and collagen cross-linking enzymes such as *lysyl oxidase (Lox)*, *Loxl2*, *Loxl3*, and *peroxidasin (Pxdn)* (Fig 7B). All these expression changes are statistically significant (log2 fold change [LFC] > 1 and adjusted *P* < 0.05.). Confirming these results, both mouse oral and pancreatic NCs contain large numbers of activated fibroblast cells and abundant collagen-based ECM (Fig 7C). In addition to the unique desmoplastic stromal response, the epithelial–mesenchymal transition (EMT) process has been closely linked to metastasis (62). The GSEA result revealed significant activation of the EMT program in mouse NCs (adjusted $P$ = 2.16 x $10^{-7}$), with strong up-regulation of mesenchymal markers including *Snai1* (63), *Zeb1* (64), and *vimentin* (*Vim*) (65, 66) (Fig 7D). Consistently, at an advanced stage, regions of both mouse oral and pancreatic NC lost the membrane-localized epithelial marker E-cadherin and gained the expression of ZEB1 and vimentin to differing degrees (Fig 7E), demonstrating EMT, which might be associated with the metastatic potential of NC. It is clear now that EMT is a reversible process marking a continuous spectrum

of phenotypes (67). However, it is under debate whether full EMT is required for metastasis and whether a reverse process called mesenchymal–epithelial transition is required for the establishment of metastasis (62). We analyzed EMT phenotypes of the metastatic lesions in mouse NC. As shown in Fig 7F, in some cases such as the lymph node metastasis of oral NC, most of the cancer cells within the metastasis tumor expressed not only E-cadherin but also vimentin and ZEB1, showing a mixed epithelial–mesenchymal phenotype. On the contrary, the mesenteric nodules developed from pancreatic NCs have lost membrane-localized E-cadherin but did not significantly up-regulate vimentin and ZEB1. Thus, there is likely a complex, non-binary interplay between desmoplasia, EMT, and metastasis of NC (more examples in Fig S8B–D). The new GEMM provides a powerful tool to dissect these interplays.

## Discussion

NUT carcinoma is an aggressive cancer with a dismal prognosis (2, 10). Because NC is extremely rare in the human population and is often diagnosed at a late disease stage, relevant in vivo models

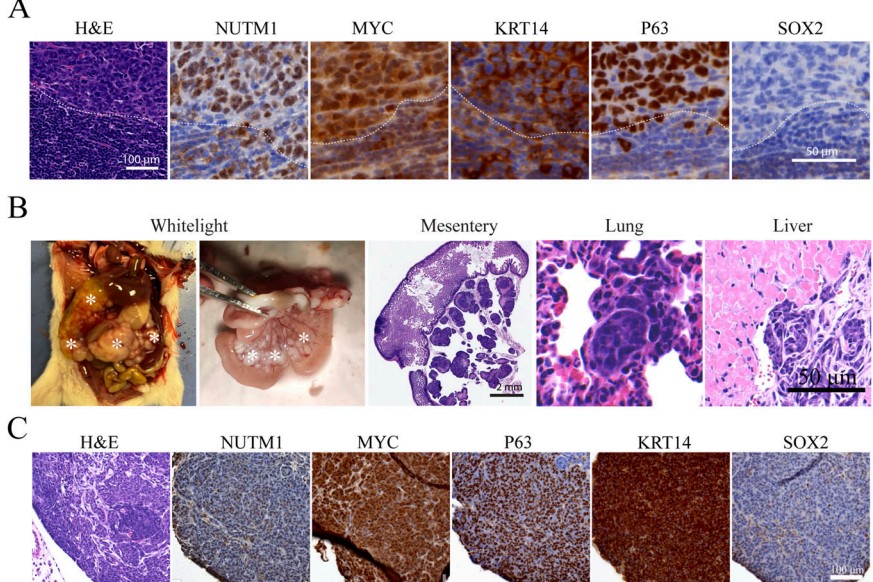

**Figure 6. mNCs are highly metastatic.**
**(A)** Representative H&E and IHC images of lymph node metastasis from oral mNC. **(B)** Representative macroscopic and H&E images of peritoneal/mesentery, lung, and liver metastasis of pancreatic mNCs. **(C)** Representative H&E and IHC images of peritoneal metastasis of pancreatic mNCs.

are vital for understanding its pathogenesis and for therapy development (20, 25). The new NC GEMM addressed key issues that are challenging to explore with currently available human cell line and xenograft models and created new opportunities for understanding NC.

First, a long-standing dilemma in NUT carcinoma research is that although the disease is strongly associated with chromosome translocation events that induce the expression of *NUTM1* fusion genes such as *BRD4::NUTM1*, the ectopic expression of the fusion genes in non-NC cells is unequivocally cytotoxic (2, 24). This dilemma raises a critical question as to whether the *NUTM1* fusion genes are indeed responsible for NC initiation. Through analyzing hundreds of NC GEMM, our data demonstrated that when expressed at the endogenous level from an endogenous chromosome translocation, *Brd4::Nutm1* can efficiently induce aggressive cancers with high penetrance. The mouse tumors that *Brd4::Nutm1* induced presented similar histopathological and molecular features of human NCs. mNCs also reproduced the intrinsic intra- and inter-tumor heterogeneity. This comprehensively characterized GEMM firmly demonstrates that the *Brd4::Nutm1* fusion gene is sufficient to induce NC-like tumors in vivo.

Second, although human NC is rare, cases have been reported in a broad range of tissues. The main affected sites were reported to be head and neck (~40% of reported cases) and thoracic (~50% of reported cases) (3). There are two possible explanations for the predilection for these sites: (1) the cell types in these tissues are more receptive to transformation by *BRD4:: NUTM1*; and (2) fewer cases were diagnosed at other sites because of limited clinical awareness. The second explanation is not trivial, considering the history of NUT carcinoma. It was first thought to be a pediatric thymic cancer (1). With the increased awareness and better availability of diagnostic methods such as the NUTM1 IHC and next-generation sequencing, NC is now considered a preferentially young adult cancer that can occur in

any age group and many tissue sites (3). Our data showed that when *Brd4::Nutm1* fusion was induced in tissue progenitors of all three germ layers using cell type–specific cre drivers, NC can be induced at 100% penetrance in the respective tissue. Third, when *Brd4::Nutm1* fusion was induced stochastically using a constitutively active Cre driver, mNCs can arise in many tissues, such as the paranasal sinuses, ovary, and kidney, previously reported in humans, and previously not reported tissues, such as the gastric tissues. Our data raised a critical possibility that the low case rate outside the head-and-neck/thoracic region might be a result of diagnosis bias, either because of limited clinical awareness or, in the case of gastric cancer, because of the low general prevalence and research intensity in North America (68). Broader and global surveys of NUT carcinoma cases in hard-to-diagnose cancers with monolithic, poorly differentiated morphology in all tissues may establish a more complete picture of NC in humans. The broad cell-type receptivity and the universal SCC phenotype at the molecular level implied that BRD4::NUTM1 coerces a shared molecular program in cell types across all germ layers to initiate NC. Thus, it appeared that *BRD4::NUTM1* has a strong reprogramming ability to drive aggressive tumors histologically consistent with a poorly differentiated SCC-like phenotype of NC regardless of the lineage context. The new GEMM provided a unique model to further investigate the key molecular mechanisms underlying NC oncogenesis.

Fourth, human NCs are highly metastatic; yet, currently, there is no established model to study this key aggressive phenotype. The GEMM uncovered an extremely strong metastatic potential of NC. GEMMs are not always optimal for studying metastasis. In many GEMMs of common cancers, mice reach their endpoint because of the growth of primary tumors before significant metastasis (69). In GEMMs that consistently develop metastasis, such as the mutant K-Ras–driven PDAC model, metastasis takes a long time to develop (15–40 wk) and is challenging to study (49, 70). The pancreatic NC

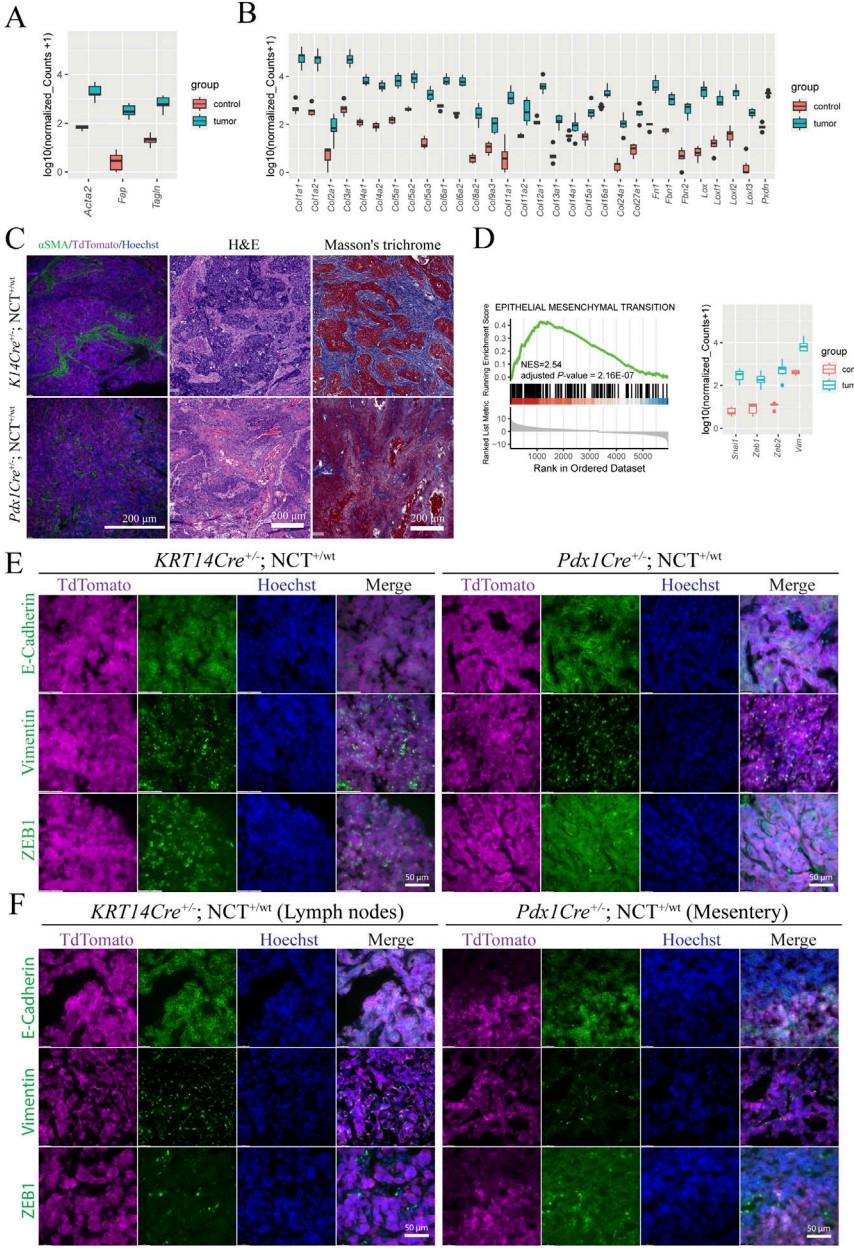

**Figure 7. mNCs present desmoplastic and EMT features that could contribute to its metastatic potential.**
**(A)** Significant up-regulation of markers of activated fibroblasts in mNCs. Data are presented as the median, and the 25th and 75th percentiles. Cutoff: abs(LFC) > 1 and adjusted *P* < 0.05. **(B)** Significant up-regulation of ECM and their modifier genes in mNCs. Data are presented as the median, and the 25th and 75th percentiles. Cutoff: abs(LFC) > 1 and adjusted *P* < 0.05. **(C)** Representative images of desmoplastic features of oral (upper) and pancreatic (bottom) mNCs, depicted by the expression of the myofibroblast marker smooth muscle alpha-actin, H&E staining, and Masson's trichrome staining of collagen-rich ECM. **(D)** GSEA enrichment of EMT genes (left) and significant up-regulation of key EMT markers in oral mNCs. Data are presented as the median, and the 25th and 75th percentiles. Cutoff: abs(LFC) > 1 and adjusted *P* < 0.05. **(E)** Expression of the epithelial marker E-cadherin and the mesenchymal markers vimentin and ZEB1 in advanced oral and pancreatic mNCs, showing regions of EMT. TdTomato staining demarcates NC cells. **(F)** Expression of the epithelial marker E-cadherin and the mesenchymal markers vimentin and ZEB1 in metastatic mNCs, presenting the heterogeneous EMT status. TdTomato staining demarcates NC cells.

GEMM develops widespread metastasis in a short time window (5–10 wk), providing ample materials to investigate the mechanisms of NC metastasis in vivo. Critically, the primary mouse NCs do present phenotypes, including desmoplasia and EMT that are associated with metastatic potentials. Further investigation using the new GEMM will lead to a fruitful understanding of the metastasis mechanism and therapeutic strategies of NC.

As immune therapy has transformed the therapeutic landscape of aggressive cancers such as melanoma and lymphoma, it is critical to investigate the immune system's interaction with targeted or immune therapies for NC. The GEMM reported here is one of the two earliest immunocompetent experimental models for NC and can provide a valuable platform for studying the immune therapies for NC.

After our preprint was online, another NC mouse model was submitted and published (71). This model uses the *cre*-dependent one-way genetic switch (FLEx) conditional inversion system and a *Sox2-CreERT2* tamoxifen driver to induce the *Brd4::NUTM1* fusion gene from the mouse *Brd4* allele. In parallel, these two GEMMs each have unique advantages and disadvantages in modeling NC. First, the FLEx model is driven by tamoxifen-inducible Cre, thus having better temporal control. However, the high recombination efficiency will mean that the fusion gene will be induced in a large number of cells simultaneously, mimicking the sequencing mutagenesis and field carcinogenesis in common cancers (72), different from the clonal stochastic reality of fusion gene–driven cancer. Our translocation model recapitulates such a reality.

Second, the FLEx model induces mNCs only in the esophagus, providing a clear and easily accessible target tissue for study. However, there has been no report of primary NC in the esophagus. Whether this inconsistency is due to species difference, mis-assigning of human primary NC, or an artifact because of pervasive induction of *Brd4::NUTM1* in all esophagus basal cells by the Sox2-CreERT remains to be determined. In contrast, our translocation model closely recapitulated the human tissue spectrum, including the rare cases outside the head-and-neck and thoracic regions, providing a richer resource understanding of the NC oncogenesis from different tissues. Furthermore, the broad tissue distribution of the translocation model will allow intersectional analysis to find the core mechanisms that drive NC and model more tissue contexts, which will be critical for future drug delivery and immune therapy research. Furthermore, because our model is a modular model based on inter-chromosomal translocation, by moving the *Brd4*-intron–hosted loxp, it is easy to construct GEMMs driven by *Nutm1* fusion genes with other partners such as *Yap1* and *Mxd4*. This level of flexibility is not offered by the FLEx model. Intriguingly, the FLEx model induced NC with a mouse *Brd4*::human *NUTM1* chimeric gene, whereas our model used the mouse *Nutm1* gene. Because the amino acid sequence is only 65% identical between human and mouse NUTM1 proteins, the observation that both fusions are oncogenic suggests that the exact sequence of NUTM1 is less critical to its oncogenic activity. Critically, NUTM1 is mainly composed of repetitive disordered sequences, which might suggest that the overall disordered nature rather than the exact sequence is required for its oncogenic function. Further analysis is warranted to clarify this problem.

In summary, we have reported the first GEMM for the NC that recapitulates the endogenous chromosome translocation syntenic to the t(15;19) that drives human NC. The new GEMM can enhance our understanding of this disease and open up many opportunities for developing targeted therapies. The GEMM will serve as a valuable preclinical tool for the global community to study the mechanism of NC and develop new treatments.

## Materials and Methods

### Ethical statement

All animal work was carried out under PROTO202000143 and PROTO202300127, approved by the Michigan State University Campus Animal Resources and Institutional Animal Care and Use Committee in AAALAC-credited facilities.

### Mouse line generation

All mouse generation work follows our published two-cell homologous recombination–CRISPR (2C-HR-CRISPR) method and protocol and is described briefly as follows (34, 73, 74). First, two-cell stage embryos of the CD1 strain were microinjected with Cas9-mSA mRNA (75 ng/µl), sgRNA targeting around the stop codon of the *Nutm1* gene (GGUGGCCCUCUGCUUCCUAC, chemically modified sgRNA from Synthego Inc., 50 ng/µl), and a biotinylated PCR donor

consisting of homology arms (5′ arm 1,000 bp and 3′ arm 600 bp) spanning the coding sequence of the *T2A-Luc2TdTomato* cassette (5 ng/µl). The *Luc2TdTomato* cassette was cloned from the pCDNA3.1(+)/Luc2-TdT plasmid (32904, a kind gift from Christopher Contag; Addgene) (35). Injected embryos were implanted into pseudopregnant females the same day to generate founder pups. Positive founders were screened by long-range PCR spanning homology arms using the following primer sets: 5′ arm gtF: GAGACTGTCATAGACAGCATCCAAGAT and 5′ arm gtR: TCATGGCTTTGTG-CAGCTGC; 3′ arm gtF: GGACTACACCATCGTGGAACAGTA and 3′ arm gtR: CATATTTAACAGTCCCACGGAGAG. A positive founder mouse was out-crossed with CD1 mice to generate N1 mice. The N1 mice were genotyped by PCR. In addition, genomic regions spanning the targeting cassette and 3′ and 5′ homology arms were Sanger-sequenced to validate correct targeting. An insertion copy number was evaluated by digital droplet PCR (ddqPCR). Heterozygous N1 mice have only one insertion copy, demonstrating single-copy insertion. An N1 that passed all quality controls was further outcrossed with CD1 for two generations to remove any potential off-target mutations introduced by CRISPR/Cas9 editing to generate N3. Heterozygous N3s were bred together to establish a homozygous colony of the *Nutm1-T2A-Luc2TdTomato* mouse line.

Then, two additional mouse lines were generated in parallel. First, a loxp site was inserted into the intron 12 of the *Brd4* gene (transcript variant 1) (*Brd4loxp*) by electroporating CD1 zygotes with Cas9 proteins, a chemically modified sgRNA (GAAUAGA-GUGCUGCUACACU; Synthego Inc.), and a single-stranded oligo-DNA donor (ssODN), with 30 base homology arms on each side spanning the loxp site (TCTCCCTGCAGAGTGGCAAGATGCCCAGTGataacttcgta-tagcatacattatacgaagttatTAGCAGCACTCTATTCTCTCCATCTGGGTA). Electroporated zygotes were implanted into pseudopregnant females the same day to generate founder pups. Positive founder pups were screened by PCR using primers: BRD4 loxp gtF: TGCCTGGGCTCTTCTGTCCAC and BRD4 loxp gtR: CAACCCAAGGTT-GATGGGACCC. A positive founder mouse was outcrossed with CD1 mice to generate N1 mice. The N1 mice were genotyped by PCR. In addition, genomic regions spanning the targeting cassette and 3′ and 5′ homology arms were Sanger-sequenced to validate correct targeting. An N1 that passed all quality controls was further out-crossed with CD1 for two generations to remove any potential off-target mutations introduced by CRISPR/Cas9 editing to generate N3. Second, a loxp site was inserted into intron 1 of the *Nutm1-T2A-Luc2TdTomato* gene (*Nutm1loxp*) by microinjecting *Nutm1-T2A-Luc2TdTomato* homozygous two-cell embryos, 75 ng/µl sgRNA targeting around the intron (ACCGGAUCCCAACCGCUCUA, chemically modified sgRNA from Synthego Inc., 50 ng/µl), and a biotinylated PCR donor consisting of homology arms (5′ arm 500 bp and 3′ arm 500 bp) spanning a loxp site and artificial genotyping primer sequences (5 ng/µl). Injected embryos were implanted into pseudopregnant females the same day to generate founder pups. Positive founders were screened by long-range PCR spanning homology arms using the following primer sets: 5′ arm gtF: GCATGAAATCAGACCAAGTGGGC and 5′ arm gtR: AAGCTGACCCT-GAAGTTCATCTG; 3′ arm gtF: GGACTACACCATCGTGGAACAGTA and 3′ arm gtR: CATATTTAACAGTCCCACGGAGAG. A positive founder mouse was outcrossed with CD1 mice to generate N1 mice. The N1 mice were genotyped by PCR. In addition, genomic regions spanning the

targeting cassette and 3′ and 5′ homology arms were Sanger-sequenced to validate correct targeting. An N1 that passed all quality controls was further outcrossed with CD1 for two generations to remove any potential off-target mutations introduced by CRISPR/Cas9 editing to generate N3. Starting from N3s, the *Brd4loxp* and *Nutm1loxp* strains were bred to double homozygous strains to generate the NCT mouse line.

For inducing the *Brd4::Nutm1* fusion, NCT$^{+/+}$ mice were crossed with heterozygous *KRT14Cre* mice (018964; Jackson Laboratory) for oral cancer, *Pdx1Cre* mice (014647; Jackson Laboratory) for pancreatic NC, *Prrx1Cre* mice (005584; Jackson Laboratory) for limb soft tissue NC, and *NLSCre* mice (3586452; MGI) for universal screen. The Cre mouse lines from Jackson Laboratory were on the B6 background and had black fur. To facilitate sensitive bioluminescence imaging, the strain was outcrossed for two generations with CD1 mice to generate albino mice. Progenies from the crosses were genotyped; the cre-positive ones were used as experimental groups, whereas the cre-negative ones were used as controls. The NCT mouse line will be available from MMRRC (MMRRC_071753-MU).

### BLI

NC mice were subjected to regular BLI to monitor tumor signals. During the imaging procedure, mice received an intraperitoneal injection of 150 mg/kg D-luciferin potassium (GoldBio). After 20 min, images of both dorsal and ventral positions were captured using IVIS Spectrum In Vivo Imaging System (PerkinElmer) under anesthesia. To quantify the tumor signal intensity, regions of interest were delineated around the head area. The total flux (photons/sec) within the regions of interest was measured using Living Image software. The average value of the total flux from both dorsal and ventral images was used for generating tumor growth curves.

### Necropsy and tissue collection

Routine necropsies were performed by ventral midline approach. Tumors, draining lymph nodes, heads, lungs, and livers were harvested and fixed in 10% neutral buffered formalin at 4°C for 24 h or flash-frozen in liquid nitrogen for further analysis. Heads were decalcified in 10% EDTA for 12–14 d at 4°C by replacing the decalcification solution every 2 d. Tissues were either routinely processed, embedded in paraffin wax, or used for cryosections.

### Histopathology

Serial, 5-$\mu$m-thick sections were cut and stained with hematoxylin–eosin (H&E). Neoplasms were characterized based on anatomic location, necrosis, inflammation (polymorphonuclear and mononuclear), and histological features (focal dysplasia, undifferentiated carcinoma, poorly differentiated carcinoma with regional squamous differentiation, and well-differentiated SCC as described in reference 75). Histopathological examination was performed by a board-certified veterinary pathologist (MFT). Histopathological diagnosis of mouse NC: carcinomas formed nests, cords, and trabeculae supported by a dense fibrous stroma with multifocal regions of necrosis. Undifferentiated carcinomas consisted of a monomorphic population of small- to intermediate-

sized basaloid cells, which were characterized by a scant amount of cytoplasm, indistinct cell borders, and round-to-ovoid nuclei with finely stippled-to-vesicular chromatin, and 0–3 prominent nucleoli. Poorly differentiated SCCs consisted of predominance of basaloid cells with focal squamous differentiation. Well-differentiated SCCs consist of multifocal regions of squamous differentiation.

### IHC

Antigen retrieval was done using sodium citrate (pH 6) for 20 min at 97°C. IHC was performed using antibodies targeting NUTM1 (1:100, HA721690; Huabio), MYC (1:500, 10828-1-AP; Proteintech), SOX2 (1:1,000, AF2018; R&D Systems), TP63 (1:1,000, AMAB91224; Sigma-Aldrich), KRT14 (1:500, NBP2-67585; Novus Biologicals), and Ki67 (1:200. NB500-170; Novus Biologicals) using Agilent/Dako Autostainer Link 48 and EnVision FLEX Kit (Dako). Immunoreactivity for each antibody was assessed by light microscopy. All antibody information was also summarized and provided in Table S2.

### Cryosection

Tissues for cryosections were dehydrated using 30% sucrose at 4°C overnight. Samples were embedded in OCT and snap-frozen on a steel bench block immersed in liquid nitrogen. Samples were sectioned at 4-$\mu$m thickness using a cryostat (Leica).

### Immunofluorescence

Slides were dried at room temperature for 30 min and permeabilized using 0.1% Tween-20 for 30 min. Samples were blocked using PBST containing 5% normal donkey serum for 1 h at room temperature. The primary antibodies diluted in blocking solution were applied on tissue sections and incubated overnight at 4°C. Slides were rinsed three times for 5 min each with PBS. Secondary antibodies diluted in blocking solution were applied on tissue sections and incubated in a dark humid chamber for 1 h at room temperature. Slides were rinsed three times for 5 min each with PBS. Nuclei were stained with Hoechst 33258 and then mounted with Vectashield mounting medium. Slides were imaged with a Leica Thunder microscope imaging system. All antibody information was summarized and provided in Table S2.

### Duo-color FISH

The Oligopaint FISH probe libraries were constructed as described previously (Xie et al, Nature Methods, 2020). A ssDNA oligo pool was ordered and synthesized as custom arrays from GenScript. Each oligo consists of a 30- to 37-nucleotide (nt) homology to the 30-kb region, flanking the mBrd4-mNutm1 fusion point from the mm10 genome assembly. Each oligo was run through OligoArray2.0 with the parameters: -n 22 -D 1,000 -l 32 -L 32 -g 52 -t 75 -T 85 -s 60 -x 60 -p 35 -P 80-m "GGGGG;CCCCC; TTTTT;AAAAA" according to algorithms developed from the Wu laboratory (https://oligopaints.hms.harvard.edu/). The mBrd4 and mNutm1 oligo subpool consists of a unique set of primer pairs for PCR amplification, a 20-nt T7 promoter sequence for in vitro transcription, and a 20-nt region for reverse transcription. The mBrd4 oligo pool was

amplified by GCGGGACGTAAGGGCAACCG and GCGTTGCGGTGCGATCTTCT. The mNutm1 was amplified by TGATAACCCACGCACGGCTG and GACCCGGGCCACTAACACGA. After PCR amplification, each Oligopaint probe was generated by in vitro transcription and reverse transcription, in which ssDNA oligos conjugated with ATTO565 and ATTO647 fluorophores were introduced during the reverse transcription step. The Oligopaint-covered genomic regions (mm10) used in this study are listed below: mBrd4, chr17:32,210,998–32,244,827; mNutm1, chr2:112,227,883–112,257,004. The oligo sequences for conjugating fluorophores during reverse transcription are as follows:/5ATTO565N/AATTCGGCAGACCCGAATGC (mNutm1) and/5ATTO647N/CCTATCCCCTGTGTGCCTTG (mBrd4).

DNA FISH is performed on cryosections prepared as described above. Tissue slides were washed twice with PBS and permeabilized in 0.5% Triton X-100 in PBS for 60 min in a Coplin jar. After washing twice in PBS, slides were treated with 0.1 M HCl for 5 min, followed by three times of washes with 2 × SSC and 30-min incubation in 2 × SSC + 0.1% Tween-20 (2 × SSCT) + 50% (vol/vol) formamide (cat#S4117; EMD Millipore). For each sample, we prepare a 25 µl hybridization mixture containing 2 × SSCT+ 50% formamide +10% dextran sulfate (cat#S4030; EMD Millipore) supplemented with 0.5 µl 10 mg/ml RNase A (cat#12091021; Thermo Fisher Scientific) +0.5 µl 10 mg/ml salmon sperm DNA (cat#15632011; Thermo Fisher Scientific) and 20 pmol probes with distinct fluorophores. The probe mixture was thoroughly mixed by vortexing and briefly microcentrifuged. The hybridization mix was transferred directly onto the slide covered with a clean coverslip. The coverslip was sealed onto the slide by adding a layer of rubber cement (Fixo gum, Marabu) around the edges. Each slide was denatured at 80°C for 5 min, then transferred to a humidified hybridization chamber, and incubated at 42°C for 16 h in a heated incubator. After hybridization, samples were washed twice for 15 min in prewarmed 2 × SSCT at 60°C and then were further incubated at 2 × SSCT for 10 min, at 0.2 × SSC for 10 min at room temperature, and at PBS for 2 × 5 min with DNA counterstaining with DAPI. Then, coverslips were mounted on slides with ProLong Diamond Antifade Mountant (Cat#P36961; Thermo Fisher Scientific) for imaging acquisition.

DNA FISH images were acquired using a structured illumination microscopy on a Zeiss Axio Observer microscope (Zeiss Elyra 7). Images were taken with a Plan-Apochromat 63×/1.40 oil DIC objective in a lens immersion medium having a refractive index of 1.515 and two pco.edge 4.2 CL HS sCMOS cameras. We used 405 nm (excitation wavelength) and 460 nm (emission wavelength) for the DAPI channel, 561 nm (excitation wavelength) and 579 nm (emission wavelength) for the ATTO565 channel, and 633 nm (excitation wavelength) and 654 nm (emission wavelength) for the ATTO647 channel. We acquired 13 phase images for each focal plane. Image post-processing was performed with ZEN 3.0 SR (black) software.

### RNA sequencing

Total RNA was prepared from tumors and normal oral mucosa tissues using RNeasy Mini Kit (QIAGEN Inc.). The quality control, cDNA library preparation (using NEBNext Ultra II RNA Library Prep Kit for Illumina), and bulk RNA sequencing were performed by Novogene Inc. following the standard protocol using the NovaSeq 6000 sequencer. Each sample was sequenced for 20 M reads.

FastQC v0.11.7 (76) was used to evaluate sequence quality, GC content, overrepresented sequences, and adapter contamination of the raw paired-end FASTQ reads for each sample. Trimmomatic v0.39 (77) was used to trim raw reads using the following settings: PE ILLUMINACLIP: TruSeq3-PE.fa:2:30:10:8:true HEADCROP:15 CROP:110. All samples passed quality control based on the results of FastQC. Quality-trimmed reads were pseudoaligned to the GENCODE mouse transcript reference (GRCm39, release M32), and transcription levels were quantified using kallisto (78) (version: 0.46.1, parameters: -b 100 -l 110). Gene expression levels were generated using tximport v1.28.0 (79) and filtered for lowly expressed genes (0.5 count per million in at least six samples). Differential expression analysis was carried out using the Bioconductor v3.17.1 (80) package DESeq2 v1.40.2 (81). rlog-normalized counts were obtained from the differential expression analysis and used for principal component analysis, sample correlation calculation, and k-means clustering (81). The Wald test was used by DESeq2 to identify genes that are differentially expressed. The Benjamini–Hochberg correction was used to adjust the P-values for multiple testing (82). The log2 fold change (LFC) was shrunken using the "apeglm" method to provide more accurate estimates (83). Differentially expressed genes (DEGs) were defined based on LFC > 1 or < –1 and adjusted P < 0.05. clusterProfiler v4.8.2 (84) was used to perform functional enrichment analysis using different databases: Gene Ontology (85, 86), KEGG PATHWAY (87), Reactome (88), and the hallmark gene sets of Molecular Signatures Database (MSigDB) (89). Lists of down-regulated DEGs and up-regulated DEGs were separately examined for overrepresentation analysis (ORA) (90). The shrunken LFCs were used to rank all DEGs decreasingly to generate a gene list as an input for GSEA (91). Redundant GO terms were removed by GOSemSim with a cutoff of 0.7 (92). Pathways or gene sets with adjusted P < 0.05 were considered significantly enriched.

For RNA fusion detection, we used STAR-Fusion v1.13.0 to call fusion events in the oral tumor and control samples using RNA-seq data with the default parameters and the developer-supplied GENCODE Mouse GRCm39 vM31 CTAT library from 09 Nov 2022 (93). Fusions were kept if fusion fragments per million total reads (FFPM) > 0.1. Fusions were filtered if fusions came from mitochondrial genes, the same gene, or paralog genes or partner genes are within 300 Kb. The R package chimeraviz v1.28.0 was used to visualize the *Brd4::Nutm1* fusion (94). RNA-seq data are available from the Gene Expression Omnibus (GEO) database as GSE263558.

### Whole-genome sequencing

Genomic DNAs were extracted from normal liver and tumor samples using the GeneJET DNA kit (Thermo Fisher Scientific), according to the manufacturer's instructions. The quality control, DNA library preparation (NEB Next Ultra II for DNA Library Prep kit), and whole-genome sequencing (WGS) were performed by Novogene Inc. following the standard protocol. A total of 1.0 µg DNA per sample was used for library preparations. Each sample was sequenced for 10–16 × coverage.

Raw reads for each sample were trimmed using Trimmomatic v0.39 (77) (PE ILLUMINACLIP:2:30:10:8:true HEADCROP:5 CROP:120). Then, the trimmed reads were aligned to the GENCODE mouse primary genome assembly (GRCm39, release M32) using the BWA-MEM algorithm in the Burrows–Wheeler Aligner (BWA) v0.7.17 (95) with the following options: -M to flag shorter split hits as secondary; -R to add read group information. The SortSam tool of Picard v2.25.0 (96) was used to sort the output files from BWA by queryname and export output as binary Sam (BAM) files. Subsequently, duplicate reads were marked using the markDuplicates tool in Picard and the resulting files were sorted using SortSam by coordinates. The AlignmentSummaryMetrics and CollectWgsMetrics tools of Picard were used to collect alignment and genome coverage statistics. The Mutect2 tool of GATK v4.1.4.1 (97) was used to identify and call variants in each sample individually. Variants were firstly filtered using FilterMutectCalls of GATK to remove probable technical or germline artifacts. Additional filters can be used to decrease the false-positive rate via SnpSift v4.1 (98): mutant allele frequency (≥5%), coverage at particular positions in tumor and normal samples (≥5×), and supporting reads for the mutation in the tumor samples (at least two). Filtered variants were further compared to exclude known polymorphisms using bcftools v1.9.64 (99). snpEff v4.1 (98) was used to annotate the effects of variants. Large structural variations were identified using Delly v0.7.8 (100) with default parameters and then filtered to get a set of confident somatic structural variants using the following sets: -f somatic -a 0.1 -m 400 -p. WGS data are available from the Gene Expression Omnibus (GEO) database as GSE263567.

### Cell-type deconvolution analysis

For the cell-type deconvolution with oral epithelium references, the signature matrixes were separately generated from two published single-cell RNA-seq datasets for implementation in the CIBERSORTx deconvolution algorithm (45). Briefly, the preprocessed count matrix of mouse tongue from 10x Genomics sequencing was extracted from the Tabula Muris. Quality control, normalization, and clustering were performed as described in the original study (46). The transcripts per kilobase million (TPM) gene expression profiles of 7,538 labeled cells, pertaining to seven distinct cell types (proliferating basal cells, basal cells, suprabasal differentiating keratinocytes, suprabasal differentiated keratinocytes, differentiated keratinocytes, filiform differentiated keratinocytes, and Langerhans cells), were used to build a signature matrix file (q value = 0.1). The number of barcode genes per cell type was held between 300 and 500, resulting in a matrix consisting of 2,256 genes. With the same strategy, the preprocessed, mouse oral epithelial basal layer cell–derived single-cell RNA-seq dataset from Jones KB et al (47) was also used to generate a signature matrix with eight cell subtypes and 1,970 genes included. The two signature matrixes were then applied to deconvolute the bulk RNA-seq TPM matrix of mNC tumors and normal controls to predict the fraction of cell subtypes, with the absolute mode, S-mode batch correction, and 100 permutations.

To correlate the cell types between the two scRNA-seq datasets, we used Seurat v4.4.0 to do a "reference-based" mapping of the query dataset (mouse oral epithelial basal layer cell–derived single-cell RNA-seq dataset) onto a reference atlas (the Tabula Muris project) (https://satijalab.org/seurat/articles/multimodal_reference_mapping) (101). In brief, after separately normalizing the two datasets by log(TPM+1), anchors between reference and query were found through FindTransferAnchors using a standard PCA transformation (dims=1:20). And then, we transferred cell-type labels from the reference to the query. Therefore, each cell in the query dataset has received an annotation defined by the reference. Cell types were validated by manually inspecting the expression of canonical markers.

### Karyotyping

Karyotyping analysis was performed on two early passage cell lines derived from mouse oral NC by KaryoLogic Inc. following the standard protocol.

### Software and statistics

The tumor growth and survival data were analyzed and visualized using Prism 9 software (GraphPad). Animal euthanasia was considered the endpoint for both analyses. The median survival times were calculated using the Kaplan–Meier analysis, and significance values were determined by conducting the log-rank test. A P-value of less than 0.05 defined statistical significance. A t test was performed for H3K27ac IF intensity.

## Data Availability

The raw and processed RNA and whole-genome sequencing data have been deposited to Gene Expression Omnibus (GEO: GSE263558 [RNA-seq]; GSE263567 [WGS]).

## Supplementary Information

## Acknowledgements

We thank Dr. Andrew Sikora, Dr. Xiao Zhao, and Dr. Jennifer Wang from the MD Anderson Cancer Center for constructive discussions. We thank MSU Transgenic and Genome Editing Facility (TGEF) for their help in generating the NC GEMM and the Histology laboratory at the Veterinary Diagnostic Laboratory (Taylor Vaughn) for their help in histopathological and IHC analysis. We would also like to thank MSU Precision Health Program Tissue Analysis Core for assistance in slide scanning. This research was supported by an R37 grant (R37CA269076) from the National Institute of Science (NIH)/ National Cancer Institute (NCI) and by startup funding both awarded to B Gu.

### Author Contributions

D Zheng: data curation, formal analysis, validation, investigation, visualization, methodology, and writing—original draft, review, and editing.

AA Elnegiry: data curation, formal analysis, validation, investigation, visualization, methodology, and writing—original draft, review, and editing.

C Luo: data curation, software, formal analysis, validation, investigation, visualization, methodology, and writing—original draft, review, and editing.

MA Bendahou: validation and methodology.

L Xie: validation, visualization, methodology, and writing—review and editing.

D Bell: formal analysis, investigation, methodology, and writing—review and editing.

Y Takahashi: formal analysis, investigation, methodology, and writing—review and editing.

E Hanna: formal analysis, investigation, and writing—review and editing.

GI Mias: software, formal analysis, validation, investigation, visualization, methodology, and writing—review and editing.

MF Tsoi: data curation, formal analysis, validation, investigation, visualization, methodology, and writing—original draft, review, and editing.

B Gu: conceptualization, resources, data curation, formal analysis, supervision, funding acquisition, validation, investigation, visualization, methodology, project administration, and writing—original draft, review, and editing.

## Conflict of Interest Statement

The authors declare that they have no conflict of interest.

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
