## [Reviewer comments · Life Science Alliance]

Life Science Alliance

Brd4::Nutm1 fusion gene initiates NUT carcinoma in vivo

Dejin Zheng, Ahmed Elnegiry, Chenxiang Luo, Mohammed Bendahou, Liangqi Xie, Diana Bell, Yoko Takahashi, Ehab Hanna, George Mias, Mayra Tsoi, and Bin Gu

DOI: <https://doi.org/10.26508/lsa.202402602>

Corresponding author(s): Bin Gu, Michigan State University and Mayra Tsoi, Michigan State University

Review Timeline:

Submission Date:	2024-01-18
Editorial Decision:	2024-02-22
Revision Received:	2024-04-09
Editorial Decision:	2024-04-26
Revision Received:	2024-04-26
Accepted:	2024-04-29

Transaction Report:

February 22, 2024

Re: Life Science Alliance manuscript #LSA-2024-02602-T

Bin Gu
Michigan State University
Obstetrics, Gynecology & Reproductive Science
3319 Bioengineering Building 775 Woodlot Dr.

Dear Dr. Gu,

Thank you for submitting your manuscript entitled "Brd4::Nutm1 fusion gene initiates NUT carcinoma in vivo" to Life Science Alliance. The manuscript was assessed by expert reviewers, whose comments are appended to this letter. We invite you to submit a revised manuscript addressing the Reviewer comments.

Thank you for this interesting contribution to Life Science Alliance. We are looking forward to receiving your revised manuscript.

Sincerely,

B. MANUSCRIPT ORGANIZATION AND FORMATTING:

Reviewer #1 (Comments to the Authors (Required)):

In the present study, Zheng and colleagues developed a novel genetically-engineered mouse model of Brd4::Nutm1-driven NUT carcinoma. Using an elegant approach of introducing cre-lox sites flanking the most common break points of the Brd4::Nutm1, they generate a Cre-inducible system to de novo generate the endogenous fusion protein in various cells of origin. Using this system, they show that endogenous expression of the Brd4::Nutm1 fusion protein results in the formation of highly aggressive NUT carcinomas in a variety of tissues.

This novel model will be a useful tool for future investigations into the biology of Brd4::Nutm1-driven carcinoma and drug testing. In general, this study is of high quality, but I have a few comments/questions.

Major comments:

- 1) Could the authors please go a bit deeper in their comparisons of the tumors caused by different Cre lines? Are there fundamental differences between different driver lines in the histomorphology, aggressiveness etc of the tumors and of the different tumor locations?
- 2) The authors generated RNA Seq data from their mouse tumors. Could they please compare the gene expression data to that of human NUT carcinomas (if available), to explore how similar the gene expression profiles between mouse and human tumors are?
- 3) Could the authors briefly characterize the immune microenvironment of some of these tumors (T cells, Macrophages, Neutrophils etc)? Eg by IHC
- 4) This may be beyond the scope of this manuscript, but the impact of this publication could be greatly improved by testing the response of the mouse tumors to BET inhibition. This could either be done in vivo or with established primary cell cultures from these mouse tumors in vitro.
- 5) Again, this may be beyond the scope of this manuscript, but the authors could use published Cut & Run data (eg GSE179692 or GSE233301) from Brd4::Nutm1 and compare the binding locations of this fusion to their mouse RNA Seq data, to determine which of the up- and down-regulated genes are direct targets of the fusion.
- 6) Do the authors have any indication if additional acquired mutations are present in these mouse tumors?

Minor comments:

- 1) In their abstract, the authors state that this is the first GEMM of NUT carcinoma. However, another model has recently been published - a fact that is also discussed in the discussion section. The authors should refrain from calling their model the first in the abstract.

Reviewer #2 (Comments to the Authors (Required)):

Review of Brd4::Nutm1 fusion gene 1 initiates NUT carcinoma in vivo by Zheng et al.

As described by the authors, Nut carcinoma (NC) stands as an aggressive malignancy lacking efficacious treatment options. Thus, the development of animal models that faithfully replicate the pathogenesis, specifically the chromosome translocation, and histopathological characteristics of human NC is of substantial importance. In this context, the current study holds significant merit as it introduces a genetically engineered mouse model that faithfully mirrors the native chromosome translocation in human, resulting in the expression of the Brd4::Nutm1 fusion gene. The study convincingly demonstrates that this fusion gene causes aggressive carcinomas in mice, exhibiting histopathological and molecular features characteristic of human NC. The utility of this model has promising potential, offering a valuable tool for mechanistic and preclinical investigations aimed at enhancing the management of this currently incurable disease.

In summary, this manuscript represents a well-executed and informative work that describes a novel mouse model of NC. The experimental methods were detailed clearly and properly conducted. The presented mouse model is relevant and state-of-the-art. While the study has merit, there are some areas needing further clarification and enhancement. The authors should explain their rationale for choosing the fusion breakpoint locations, provide more detailed depiction of the result data, and include appropriate experimental controls among other suggestions. The followings are specific comments that the authors may wish to consider:

Major Comments:

1. In this study, loxp sites were inserted into intron 10 of the mouse *Brd4* gene and intron 1 of mouse *Nutm1* gene, resulting in fusion of *Nutm1* starting from exon 2 downstream to *Brd4* exon 10. This differs from another NC mouse model (Durall et al., 2023) in which the breakpoint is located in the intron 16 of *Brd4* gene. Additionally, unlike the NC model developed by Durall et al (Durall et al., 2023). in which human *Nutm1* was cloned downstream to mouse *Brd4*, this study induced endogenous chromosomal translocation and expression of the native mouse *Brd4::Nutm1* fusion gene. Since this is a novel NC mouse model, the authors should expand their discussion on the rationale behind choosing the fusion breakpoint for the *Brd4::Nutm1* gene in mice. In particular, they should address the sequence similarities with fusion variants found in human NC patients, and the functional protein domains critical for oncogenesis of the fusion gene. Human *NUTM1* and Mouse *Nutm1* are only ~65% identical. What do the findings of this study suggest about the importance (or unimportance) of specific domains of *Nutm1*?
2. Many of the presented data and figures lacked sufficient descriptive details, making it challenging to comprehend the information they convey and the purpose they serve. This significant deficiency needs to be addressed by providing more thorough depictions of the data and figures in the Result section.
In Figure 1F, FISH from what appears to be a single cell is shown. Please provide more examples, either in the main figure or as a supplement.
In Fig. 1G, without any quantification data, it is unclear how the authors confirmed the low frequency and stochastic nature of the chromosome translocation by presenting figures showing *Nutm1*-positive cells surrounded by normal epithelial or tumor-associated stromal cells.
The description of Fig. 2A is vague. It would improve the clarity to use arrows or other indicators to highlight areas of basaloid, polygonal, or spindle-shaped cells, as well as areas of squamous cell differentiation and keratinization, respectively. Insets with higher magnification would be helpful. Regarding Fig. 2B, it is unclear what criteria or references the pathologist used to evaluate the cell differentiation level. The methods reference another publication however that paper does not spell out the histopathologic criteria used. As for Fig. 2C, the authors should further clarify the meaning of "variable morphology." Providing separate highlights and detailed depictions of variable morphology in Fig. 2C would be beneficial.
Figure 2E nicely shows differences in H3K27Ac staining in tumor and non-tumor tissue. However only a very small area of stromal/non-tumor tissue is shown. The figure would be more convincing with a larger field of view or additional examples. Again, quantification is needed here.
3. Several studies have reported KRT14 as a canonical marker specific to squamous differentiation (Durall et al., 2023; Prall et al., 2021; Zhang et al., 2020). However, the data presented showed high KRT14 expression in the tumors in this NC mouse model, which appears inconsistent with the authors' statement that the tumor cells were highly undifferentiated in this NC mouse model. Please address this apparent discrepancy.
4. Lack of essential controls: Conducting staining on comparable anatomical locations of WT mice would offer insights into the expression level and location of tumor-related genes in normal tissue. This comparison is particularly crucial for assessing genes with low expression in tumor tissue, such as *Sox2*, to verify that the low expression observed in the tumor tissue remains significant.
5. Western blot analysis of *Nutm1*, demonstrating bands of the expected size, could also be employed to further validate the expression of the *Brd4::Nutm1* fusion gene.
6. Throughout the paper, the data were primarily descriptive, with a notable absence of quantification for the histological findings. Although quantification of histological data can be challenging, alternative approaches such as western blotting could be considered. For instance, determining whether the expression level of *Brd4::Nutm1* correlates with the expression levels of P63, cMyc, or H3K27ac would offer valuable mechanistic insights and enhance the informativeness of the findings.
7. Tests of statistical significance are missing from figure panels, for example Fig. 3C, Fig. 7 A,B
8. One of the most significant, and surprising, findings of the manuscript is the appearance of *Brd4-Nutm1*-driven squamous cell carcinomas in multiple tissues, including those of non-ectodermal germ layer origin. This finding warrants some more discussion. The authors make some fairly broad claims about reprogramming, for example lines 171-182 and 230-232. However, the transcriptome analysis (Fig. 3B) is superficial and highlights hallmark pathways associated with cancer in general, and is not informative about lineage-specific differentiation mechanisms. To identify the developmental processes driving squamous cell differentiation from diverse lineages in this NC mouse model, tumors could be harvested at an earlier time point, prior to the development of multiple and massive tumors. This is, in my opinion, beyond what should be required for this first manuscript. But if the authors do not undertake this type of experiment, they should temper their conclusions, for example acknowledging that *Cre* drivers may not be completely lineage-specific and that it is difficult to rule out any unanticipated expression of the fusion oncogene in rare cell types. Similarly, please comment on possible variable/mosaic recombination driven by NLS-*Cre*. Given the cellular toxicity of *BRD4-NUTM1*, it is surprising that mice viably develop in this model, and survive to form tumors. What accounts for the (relative) selectivity of tumor developing in this model-is there some sort of selection of which tissue types actually go on to form tumors?

9. The claim that morphologic heterogeneity explains plasticity/treatment resistance is highly speculative and not supported by direct evidence. Bromodomain inhibitors (BETi) have demonstrated efficacy in suppressing tumor growth and extending survival in both animal models and human NC trials. The NC mice could be treated with BETi to assess its potential as a preclinical model for developing new treatments (Durall et al., 2023), and to directly test the idea that different cell morphologies will be associated with selective treatment resistance.

Minor Comments

1. Fig. 1F: Including and analyzing low-power DC-FISH images, particularly in younger or fetal samples, could offer valuable insights into translocation frequency.
2. Fig. 1G: In panel a, the highlighted regions meant to correspond to panel b and c were erroneously labelled as "a" and "b"
3. Supplementary Fig. 2A: No information about the location or tissue type associated with this image was provided.
- Supplementary Fig. 2C: For improved clarity, consider adding a box to the low-magnification image to highlight the exact location corresponding to the zoomed-in images.
4. Line 245: The correct genotype, NLS-Cre, was mistakenly typed as "NLC-Cre" in the text.
5. In NLS-Cre transgenic mice, did individual animals develop multiple tumors across different organs?
6. Fig. 5D: Was Alcian Blue also performed on the Pdx1Cre;NCT samples?
7. Fig. 1C: Fig. 1C seems to be missing a description in the Results section.

Durall, R. T., Huang, J., Wojenski, L., Huang, Y., Gokhale, P. C., Leeper, B. A., . . . French, C. A. (2023). The BRD4-NUT Fusion Alone Drives Malignant Transformation of NUT Carcinoma. *Cancer Res*, 83(23), 3846-3860. <https://doi.org/10.1158/0008-5472.CAN-23-2545>

Prall, O. W. J., Thio, N., Yerneni, S., Kumar, B., & McEvoy, C. R. (2021). A NUT carcinoma lacking squamous differentiation and expressing TTF1. *Pathology*, 53(5), 663-666. <https://doi.org/10.1016/j.pathol.2020.09.027>

Zhang, X., Zegar, T., Lucas, A., Morrison-Smith, C., Knox, T., French, C. A., . . . Siveke, J. T. (2020). Therapeutic targeting of p300/CBP HAT domain for the treatment of NUT midline carcinoma. *Oncogene*, 39(24), 4770-4779. <https://doi.org/10.1038/s41388-020-1301-9>

Reviewer #3 (Comments to the Authors (Required)):

Summary:

This paper by Zheng and coworkers describes the establishment of an in vivo mouse model of NUT carcinoma (NC), driven in humans mostly by the BRD4::NUTM1 gene fusion. The authors elegantly describe how the model has been engineered allowing for the induction of the chromosomal translocation resulting in the fusion. This is especially important since ectopic expression of the fusion is toxic in non-NC cell lines. By allowing for a more natural expression level due to the chromosomal translocation, a more representative model of human NUT carcinoma can be established. Using several methods, the authors show that the model can recapitulate human NUT carcinoma in mice, both in site of tumour origin and histomorphologically. This will offer a great tool for studying this disease further and developing potential novel treatments in the future. One other model has already been recently published, but the authors attempt to describe why their model may be better suited to study NC.

Main points:

Model development and validation - data is supportive

Histopathology - data is supportive

Transcriptomic investigations - data is supportive

Other comments:

Major comments:

1. Regarding the validation of the t(2;17) in mice - the authors should consider showing the output given by Delly as a supplementary table. Was the breakpoint consistent with expectations (correct location in the introns of Brd4 and Nutm1?) On what material was Sanger sequencing performed to finally validate the fusion? Are any unexpected alternative splice variants seen in RNA data? If the authors have RNA-sequenced some of the tumours, a fusion calling algorithm can be used to also check for this.
2. In the introduction the authors state that the fusion is thought to be enough to drive human HC in an otherwise quiet genomic background. Upon WGS of the two shown tumours in supplementary Figure 1, the circos plots show anything but a quiet genome. What cutoffs were used for SVs in Delly and how were these further filtered? Have the authors considering using a second variant caller and merging the output with that of Delly? Any thoughts on why so many SNVs are detected? Is this seen in human NC as well?
3. In both figure 2 and supplementary figure 2, the authors show IHC of known highly expressed proteins in NC such as MYC, TP63 and so on. Could the authors correlate this with RNA expression data for these genes? Does Sox2 show a lower expression for instance in bulk RNA if only selected regions show a high protein expression? Additionally, there is likely an error in the legend of Figure 2E - the authors refer to (b) and (c) when there are only (a) and (b) in the figure.
4. The authors show that many genes are up and downregulated in mouse NC vs. normal cells. Is there any published expression data from human NCs to correlate with? Are the same genes or pathways rather up and down regulated in human

NC as well? Providing a list of these genes as a supplementary table may be helpful for future comparison studies. Additionally, the deconvoluted bulk RNA data shown in supplementary figure 3D should be further clarified. Which annotations in that set match those (even if not exactly) from figure 3C.

5. The text around that part of the discussion discussing the model developed by Dural et al. should be updated to portray that this model is not the first immunocompetent one, while still lifting the strengths of this model.

Minor comments:

1. The authors should check for consistency in using :: to denote gene fusions as this is interchanged with the older standard of using a dash, especially in the introduction. The same applies to double checking italics style for genes and normal font for proteins and how human or mouse genes are denoted to make it clearer for the reader. An example is found in the first subtitle in the results section 3.1 - do the authors mean the human fusion or the mouse one here?
2. Some sentences are poorly written or too long and not easy to understand (such as lines 55-56 in the introduction or 108-112 in the results). The authors should consider correcting and rewriting them in a clearer way.
3. A clarification for which cells/tissues are used for each of the shown IF/IHC results in the figures would be helpful for readers who are not used to performing such stainings.
4. BLI should be spelled out the first time as not all readers may be aware of what it stands for.
5. Genes shown to be upregulated in Figure 7A-B should be italicised. Additionally, are similar genes known to be upregulated in human NC (similarly to my previous comment)?
6. What secondary antibodies were used for immunofluorescence? This is not clear in the methods section.
7. Which kit were used for cDNA library preparation for RNA sequencing?
8. There is an error in section 4.10 - The title described WGS but the text partially describes RNA sequencing.

Dear Reviewers,

First, we would like to thank you for your time and effort spent reviewing our manuscript on developing a novel NUT Carcinoma GEMM. We thank you for the overall enthusiasm and generally positive comments, and all the very constructive suggestions. We acknowledge that it is still the early days for using GEMMs to understand NUT carcinoma and, in fact, the early days of developing a molecular and developmental understanding of NUT carcinoma. Significant nuance remains for the characterization of human NC and mouse NC models. In the revision, we aim to address most of your comments and suggestions to the best of our ability. However, for some comments, such as those suggesting comparing our transcriptomics data with human data, we need to acknowledge the scarcity of human data and the very different circumstances in which those data were collected. We felt that a clear comparison with appropriate statistical power might not be achieved in those cases. Thus, any conclusion from such comparisons might not be well-founded and possibly misleading. We will explain the details of our rationales in cases where we cannot address the suggestion in our point-by-point responses. The main purpose of this manuscript is to describe this important genetic model and quickly make it available to the broad research community. We believe that is the best way to clear up all the mysteries and nuances around NC as a community. Thanks again for your efforts; we look forward to your positive response.

Reviewer #1 (Comments to the Authors (Required)):

In the present study, Zheng and colleagues developed a novel genetically-engineered mouse model of Brd4::Nutm1-driven NUT carcinoma. Using an elegant approach of introducing cre-lox sites flanking the most common break points of the Brd4::Nutm1, they generate a Cre-inducible system to de novo generate the endogenous fusion protein in various cells of origin. Using this system, they show that endogenous expression of the Brd4::Nutm1 fusion protein results in the formation of highly aggressive NUT carcinomas in a variety of tissues.

This novel model will be a useful tool for future investigations into the biology of Brd4::Nutm1-driven carcinoma and drug testing. In general, this study is of high quality, but I have a few comments/questions.

Major comments:

1) Could the authors please go a bit deeper in their comparisons of the tumors caused by different Cre lines? Are there fundamental differences between different driver lines in the histomorphology, aggressiveness etc of the tumors and of the different tumor locations?

Thanks so much for your kind suggestion of comparing the tumor developed from different cre drivers. Human NUT carcinoma presents a highly variable phenotype. They are mostly characterized as poorly differentiated carcinoma with or without abrupt keratinization. However, the phenotype itself is highly variable. And as evidenced by various clinical reports (e.g. doi: 10.1002/hed.27553 (12 cases of sino-nasal NCs), 10.1097/PAS.0000000000001967 (14 cases of Thyroid NCs), 0.1111/his.14306 (6 cases of lung NCs), there is a high level of variability regarding the histology features and marker expression. And there was no clear correlation between the anatomical site of the tumors and their histological and marker expression presentation. With the limited number of cases we have analyzed in our mouse GEMM from different organs and tissues, driven by different cre, we saw a similar variable characteristic of tumors. We felt that any conclusion about the similarity or difference according to the anatomical or lineage origin of mouse NC would be premature at this point and might provide some misleading information, which might need to be corrected later. Thus, we made it clear in our abstract, lines 58-60. 'Similar to the reports of human NC incidence, *Brd4::Nutm1* can induce NC from a broad range of tissues with a strong phenotypical variability.'

2) The authors generated RNA Seq data from their mouse tumors. Could they please compare the gene expression data to that of human NUT carcinomas (if available), to explore how similar the gene expression profiles between mouse and human tumors are?

Thanks for your kind suggestion of comparing our RNA seq data with human NC RNA seq data. At this point, only RNAseq data on one human primary NCs are available in the public domain (Durall et al., 2023) (the paper described two but only the 28284 pleural tumor is on GEO). All the other human NC RNAseq data are on cell lines. Also like many human datas, the appropriate normal tissue control data is unavailable. Considering that our RNAseq data is collected from NC derived from oral mucosa (6 tumor and 6 normal oral mucosa copntrols) and that BRD4::NUTM1 regulated gene program is likely to be impacted by the gene expression and epigenetic status of their cell of origin (Alekseyenko et al. 2015, [10.1101/gad.267583.115](https://doi.org/10.1101/gad.267583.115)), we do not think it would be appropriate comparison with our tumors and the scarce information on human NC (which will lead to lack of statistical power). From our IHC and IF study, we do find the expression of NC markers such as P63, cMYC and SOX2, the RNAseq data confirmed them. Other than that, we do not feel that the comparison will lead to any meaningful conclusion. In fact, if we put forward any conclusion based on such a comparison, it would be premature and potentially misleading. Thus, we believe it would be best if we left this comparison to a future, more stringent study when we collected appropriate numbers of RNAseq data from human primary NCs, as well as appropriate normal tissue controls, which we are working towards.

3) Could the authors briefly characterize the immune microenvironment of some of these tumors (T cells, Macrophages, Neutrophils etc)? Eg by IHC

Thanks for the suggestion. We performed some immune cell stain in our mouse NC model and had a few pictures on the first version of our preprint (<https://www.biorxiv.org/content/10.1101/2023.07.29.551125v1>). However, upon staining more samples, we found that the immune infiltration phenotype is also quite variable in mouse NCs. Thus, we do not feel that a few representative pictures will represent the immune phenotype of NCs comprehensively. To address this question, we have also performed an immune cell profiling done by CIBERSORT of our RNA-seq data. Again, the results revealed high variability between tumors (results attached in this response). We again believe any conclusion on the published paper would be premature and potentially misleading. Further analysis with more samples using methods such as flow cytometry and single-cell RNAseq will further characterize the immune phenotype of mouse NC, which would be main topic for a future paper.

[Figure removed by editorial staff per authors' request].

4) This may be beyond the scope of this manuscript, but the impact of this publication could be greatly improved by testing the response of the mouse tumors to BET inhibition. This could

either be done in vivo or with established primary cell cultures from these mouse tumors in vitro.

Thanks for the suggestion. We performed a test of BETi JQ1 treatment on the mNC cell line we derived, and the cells were sensitive to BETi (attached a picture here). We did not investigate this further because, as heavily investigated in human NC xenograft models and patients, the effectiveness of BETi is highly variable between patients and short-lived. This short-lived effect with mild life extension was also found in the other NC GEMM paper ((Durall et al., 2023) using ABBV-744 BET inhibitor. With limited resources, we decided not to pursue redundant efforts on developing BETi for NC therapy and look for other therapeutic avenues. hopefully, we can share it in another paper.

5) Again, this may be beyond the scope of this manuscript, but the authors could use published Cut & Run data (eg GSE179692 or GSE233301) from Brd4::Nutm1 and compare the binding locations of this fusion to their mouse RNA Seq data, to determine which of the up- and down-regulated genes are direct targets of the fusion.

Thanks for the suggestion. We checked and analyzed these data. However, the GSE179692 is on the TC-797 pleural NC cell line, which is of a different cell type of origin from our oral mucosa-derived NC used for RNA-seq. As suggested by (Alekseyenko et al. 2015, [10.1101/gad.267583.115](https://doi.org/10.1101/gad.267583.115)), the binding pattern of BRD4::NUTM1 is cell type dependent. We did detect the expression of classical NC markers such as cMYC, P63 that was shown to be bound by BRD4::NUTM1 in GSE179692. But more than that, we don't feel that we can perform a comparison that lead to accurate and prudent conclusion. As for GSE233301, that was on HEK293 cells overexpressing BRD4::NUTM1, which is even further with regard to cell lineage from our tumors. For the same reason, we don't feel it is a good comparison. (Durall et al., 2023) did publish a BRD4::NUTM1 CUT&RUN data on their mouse NC. However, upon analyzing their raw data, this CUT&RUN dataset seems to represent a failed CUT&RUN experiment without any enrichment for chromatin regions, essentially background signals everywhere. Thus, we cannot do the comparison. We regret that we cannot address this question but felt that it's prudent not to make any conclusions rather than make conclusions that we are not confident of.

6) Do the authors have any indication if additional acquired mutations are present in these mouse tumors?

Although the published reviews often state that BRD4::NUTM1 drives NC alone in a quiet genomic landscape, there are not many genomic sequencing analyses to support that conclusion. To address this question, I looked through all case reports on PubMed about Nut carcinoma and found three studies that reported tumor mutation burden (mostly analyzed by comprehensive genomic profiling panels, with one case with Whole Genome Sequencing) of a total of 10 cases of human NUT carcinoma (Doi: 10.1186/s13023-020-01449-x; 10.1093/annonc/mdw686; 10.21037/acr-19-168). These studies reported a tumor mutation burden of >1.1 mutation/Mbs all the way to 73.81 mutations/Mbs, demonstrating a widely variable mutation burden in human NC. All three studies stated that there were no apparent

functional mutations of tumor suppressors or well-known oncogenes discovered. However, since there are no information on the exact mutation that was discovered in each cases, we cannot determine if there are any recurrent mutations.

In our manuscript, we briefly described two pairs of WGS analyses, both from oral tumor and normal liver tissue pairs from the same mouse. The mutation burden on both tumors is 2.6 mutations/Mbs and 3.1 mutations/Mbs, respectively. We did not discover any functional mutations in tumor suppressor or oncogenes or any recurrent mutations or structural variance other than the t(2;17) chromosome translocation we induced. However, this does not mean we will not detect any additional functionally important mutations that are recurrent in some subset of NCs once we analyzed more samples in the future. We added the mutation calling methods in the method section but felt that any conclusion about the mutation load would be premature to include.

Minor comments:

1) In their abstract, the authors state that this is the first GEMM of NUT carcinoma. However, another model has recently been published - a fact that is also discussed in the discussion section. The authors should refrain from calling their model the first in the abstract.

Thanks for reminding us about the other GEMM. We publicly announced our model, including all the detailed design and the characterization of the oral NCs in the first version of our preprint on July 30th, 2023. According to the metadata published by Durall et al. 2023, they submitted their paper for review on August 24th, 2023 to Cancer Research and got published through the accelerated priority report on Oct 8th, 2023. Although we are well aware that they knew about our preprint, Durall et al. made no mention of our model and claimed they made the first NC GEMM. Thus, chronologically, we are the first to report an NC GEMM in the public domain.

That being said, I am aware that right now, there is still some grey zone on recognizing preprint, which is highly dependent on individual opinion (though it should not have been), I modified the statement in the abstract to (lines 53-55) *'Here, we report the first genetically engineered mouse model (GEMM) for NUT carcinoma that recapitulates the t(15;19) chromosome translocation in mice.'* We believe this is an accurate statement because our model is the first to induce the endogenous syntenic chromosome translocation, whereas the one reported by Durall et al. used FLEx method to ectopically activate a mouse Brd4-human NUTM1 chimeric gene from the *Brd4* locus.

Reviewer #2 (Comments to the Authors (Required)):

Review of Brd4::Nutm1 fusion gene 1 initiates NUT carcinoma in vivo by Zheng et al. As described by the authors, Nut carcinoma (NC) stands as an aggressive malignancy lacking efficacious treatment options. Thus, the development of animal models that faithfully replicate the pathogenesis, specifically the chromosome translocation, and histopathological characteristics of human NC is of substantial importance. In this context, the current study holds significant merit as it introduces a genetically engineered mouse model that faithfully mirrors the native chromosome translocation in human, resulting in the expression of the Brd4::Nutm1 fusion gene. The study convincingly demonstrates that this fusion gene causes aggressive carcinomas in mice, exhibiting histopathological and molecular features characteristic of human NC. The utility of this model has promising potential, offering a valuable tool for mechanistic and preclinical investigations aimed at enhancing the management of this currently incurable disease.

In summary, this manuscript represents a well-executed and informative work that describes a novel mouse model of NC. The experimental methods were detailed clearly and properly conducted. The presented mouse model is relevant and state-of-the art. While the study has

merit, there are some areas needing further clarification and enhancement. The authors should explain their rationale for choosing the fusion breakpoint locations, provide more detailed depiction of the result data, and include appropriate experimental controls among other suggestions. The followings are specific comments that the authors may wish to consider:

Major Comments:

1. In this study, loxp sites were inserted into intron 10 of the mouse *Brd4* gene and intron 1 of mouse *Nutm1* gene, resulting in fusion of *Nutm1* starting from exon 2 downstream to *Brd4* exon 10. This differs from another NC mouse model (Durall et al., 2023) in which the breakpoint is located in the intron 16 of *Brd4* gene. Additionally, unlike the NC model developed by Durall et al (Durall et al., 2023). in which human *Nutm1* was cloned downstream to mouse *Brd4*, this study induced endogenous chromosomal translocation and expression of the native mouse *Brd4::Nutm1* fusion gene.

Since this is a novel NC mouse model, the authors should expand their discussion on the rationale behind choosing the fusion breakpoint for the *Brd4::Nutm1* gene in mice. In particular, they should address the sequence similarities with fusion variants found in human NC patients, and the functional protein domains critical for oncogenesis of the fusion gene. Human *NUTM1* and Mouse *Nutm1* are only ~65% identical. What do the findings of this study suggest about the importance (or unimportance) of specific domains of *Nutm1*?

Thanks so much for pointing out the discrepancy between the description of the fusion structure. Our fusion design recapitulated the exact structure and junction sequence reported by the French group in 2003 (PMID: 12543779). We apologize that we have made a mistake in counting the exon number due to missing the first two non-coding exon of *Brd4* in our original manuscript. Now the description has been changed to intron 12 in both main text and Figure 1. This fusion structure is also a recurrent one reported in most NC cases. To resolve the discrepancy with the Durall paper, we localized their fusion junction by analyzing their genotyping primers (the exact DNA sequence of their KI construct was not provided). We found that their junction intron is exactly the same one as ours (an alignment attached with this response), thus, it's possible that they miscounted their intron number and wrote intron 16.

The question about the 65% identity of the human and mouse NUTM1 protein and why they both can drive NC is a really important and interesting one. We speculate that this is due to the fact that NUTM1 is largely composed of disordered sequences that engage in multivalent weak interactions. As a result, the selection constraint on NUTM1 is not on the exact protein sequence, but the general disordered nature, which has been conserved across species. Further future studies are needed to validate this speculation. We added a passage in discussion after we discuss the Durall et al. model about this line 546-552 .

2. Many of the presented data and figures lacked sufficient descriptive details, making it challenging to comprehend the information they convey and the purpose they serve. This significant deficiency needs to be addressed by providing more thorough depictions of the data and figures in the Result section.

In Figure 1F, FISH from what appears to be a single cell is shown. Please provide more examples, either in the main figure or as a supplement.

In Fig. 1G, without any quantification data, it is unclear how the authors confirmed the low frequency and stochastic nature of the chromosome translocation by presenting figures showing Nutm1-positive cells surrounded by normal epithelial or tumor-associated stromal cells.

The description of Fig. 2A is vague. It would improve the clarity to use arrows or other indicators

to highlight areas of basaloid, polygonal, or spindle-shaped cells, as well as areas of squamous cell differentiation and keratinization, respectively. Insets with higher magnification would be helpful. Regarding Fig. 2B, it is unclear what criteria or references the pathologist used to evaluate the cell differentiation level. The methods reference another publication however that paper does not spell out the histopathologic criteria used. As for Fig. 2C, the authors should further clarify the meaning of "variable morphology." Providing separate highlights and detailed depictions of variable morphology in Fig. 2C would be beneficial.

Figure 2E nicely shows differences in H3K27Ac staining in tumor and non-tumor tissue.

However only a very small area of stromal/non-tumor tissue is shown. The figure would be more convincing with a larger field of view or additional examples. Again, quantification is needed here.

Thanks for pointing out these deficiencies, we have addressed them one by one here:

- In Figure 1F, FISH from what appears to be a single cell is shown. Please provide more examples, either in the main figure or as a supplement.

We now provided a low magnification figure in Figure S2A.

- In Fig. 1G, without any quantification data, it is unclear how the authors confirmed the low frequency and stochastic nature of the chromosome translocation by presenting figures showing Nutm1-positive cells surrounded by normal epithelial or tumor-associated stromal cells.

We have now made it clear that the left panel of Fig. 1G is a picture of the normal oral mucosa region of an NC-bearing KRT14Cre;NCT^{+/-} mouse. Since a large proportion of these oral mucosa cells are Krt14Cre positive, but we did not detect any isolated NUTM1 positive cells, this indicates that the rate of chromosome translocation that form Brd4::Nutm1 is low. Furthermore, we included more NUTM1 IHC images of normal oral mucosa regions of NC-bearing mice and performed cell count in Figure S3. These data demonstrate that in thousands of normal oral mucosa cells, there is not a single isolated NUTM1-positive cell. This indicates that the frequency of translocation is low, and the few that got the translocation developed into tumors. We described these results at line 181-185.

- The description of Fig. 2A is vague. It would improve the clarity to use arrows or other indicators to highlight areas of basaloid, polygonal, or spindle-shaped cells, as well as areas of squamous cell differentiation and keratinization, respectively. Insets with higher magnification would be helpful. Regarding Fig. 2B, it is unclear what criteria or references the pathologist used to evaluate the cell differentiation level. The methods reference another publication however that paper does not spell out the histopathologic criteria used. As for Fig. 2C, the authors should further clarify the meaning of "variable morphology." Providing separate highlights and detailed depictions of variable morphology in Fig. 2C would be beneficial.

We have added a supplementary panel Figure S4A to illustrate the representative morphologies and added explanations in main text. Regarding Fig 2B, we have added the description in the method section line 664-672. For Figure 2C, we have clarified in the main text that '*Notably, even within a single mouse, individual tumors can display variable levels of keratinization and stroma infiltration (Figure 2C)*' (line 212-213).

- Figure 2E nicely shows differences in H3K27Ac staining in tumor and non-tumor tissue. However only a very small area of stromal/non-tumor tissue is shown. The figure would be more convincing with a larger field of view or additional examples. Again, quantification is needed here.

We have provided more staining pictures on non-tumor cell regions in Figure S4C and quantification of H3K27ac fluorescence intensity in Figure S4D.

3. Several studies have reported KRT14 as a canonical marker specific to squamous differentiation (Durall et al., 2023; Prall et al., 2021; Zhang et al., 2020). However, the data presented showed high KRT14 expression in the tumors in this NC mouse model, which appears inconsistent with the authors' statement that the tumor cells were highly undifferentiated in this NC mouse model. Please address this apparent discrepancy.

Thanks for pointing out the discrepancy about the KRT14 marker. KRT14 is a broadly acknowledged marker of the basal progenitor cells of oral mucosa, esophagus and respiratory epithelial (doi: [10.1016/j.stem.2018.10.015](https://doi.org/10.1016/j.stem.2018.10.015); [10.1016/j.tcb.2018.04.007](https://doi.org/10.1016/j.tcb.2018.04.007); [10.1038/s41556-021-00679-w](https://doi.org/10.1038/s41556-021-00679-w); [10.1172/jci.insight.162041](https://doi.org/10.1172/jci.insight.162041)), as well as other epithelial tissues such as bladder epithelial (doi: [10.1038/ncomms11914](https://doi.org/10.1038/ncomms11914)). And Krt14cre and creERT driver has been broadly accepted as specific cre for basal progenitor/stem cells of oral mucosa and airway (doi: [10.1242/dev.127.22.4775](https://doi.org/10.1242/dev.127.22.4775); [10.1073/pnas.0906850106](https://doi.org/10.1073/pnas.0906850106)). Prall et al. 2021 was talking about an atypical NC case that was negative for Krt14, which should not be the reason to disqualify Krt14 as a basal progenitor marker. Zhang et al., 2020 were talking about skin, which has a relatively different cellular hierarchy and marker groups as the internal organ epithelium. For example, the skin basal cell does not express SOX2, whereas the oral mucosa basal cell does not. Perhaps the situation in the skin should not be used to understand the NC we derived from oral mucosa and other internal organs. The Durall et al., 2023, stated that Krt14 and Krt5 are squamous differentiation-specific markers but did not provide any citation to support their claim. In fact, Krt5 is an even more established and slightly more specific basal progenitor marker for epithelial tissues (doi: [10.1073/pnas.0906850106](https://doi.org/10.1073/pnas.0906850106); [10.1016/j.stem.2018.10.015](https://doi.org/10.1016/j.stem.2018.10.015)). The experiments they described using Krt14 and Krt5 are cell line studies that may or may not recapitulate in vivo conditions. Thus, we believe the expression of Krt14 can be used as one marker for our mouse NC's undifferentiated phenotype. The undifferentiated status was determined by a combination of morphological analysis, marker expression such as TP63, KRT14, and clustered SOX2, as well as the CIBERSORT analysis of the RNAseq data, which showed an enrichment of basal progenitor phenotype.

4. Lack of essential controls: Conducting staining on comparable anatomical locations of WT mice would offer insights into the expression level and location of tumor-related genes in normal tissue. This comparison is particularly crucial for assessing genes with low expression in tumor tissue, such as Sox2, to verify that the low expression observed in the tumor tissue remains significant.

Thanks for pointing this out. We have added stainings of normal tissues in regions at a similar anatomical location in NC-bearing mice to all IHC panels (Figure S4E, Figure S6 B,D and Figure S7D).

5. Western blot analysis of Nutm1, demonstrating bands of the expected size, could also be employed to further validate the expression of the Brd4::Nutm1 fusion gene. We have included it in Figure S2H.

6. Throughout the paper, the data were primarily descriptive, with a notable absence of quantification for the histological findings. Although quantification of histological data can be challenging, alternative approaches such as western blotting could be considered. For instance, determining whether the expression level of Brd4::Nutm1 correlates with the expression levels of P63, cMyc, or H3K27ac would offer valuable mechanistic insights and enhance the informativeness of the findings.

Thanks for pointing this out. The level of all these markers shows high heterogeneity and variability both from case to case and at the cell population level both in reported human cases (there are cases with strong BRD4::NUTM1 expression but no P63 expression: (doi: [10.1186/s13000-020-01053-4](https://doi.org/10.1186/s13000-020-01053-4))) and mouse tumors (shown by our IHC). It is not likely a simple linear correlation can be established at the bulk tissue level. In addition, Western Blotting is only a semi-quantitative method that may not be reliable in detecting subtle differences in the levels of these markers. Thus, WB in a few tumors might not be sufficient to derive any accurate conclusions on the correlation between BRD4::NUTM1 expression and the expression of these markers.

To address this question, we performed a quantitative correlation experiment with the RNA sequencing data, which is more quantitative than western blotting. We showed the data in Figure S4F and added explanations in the main text line '*To understand if the observed variation of the expression of key NC markers, including Myc and Trp63, and the variation in the Sox2-expressing cell population is related to the expression level of the Brd4::Nutm1 fusion gene, we analyzed their expression level using the quantitative bulk RNA-sequencing data (discussed in more details below). As shown in Figure S4F, although the expression levels of Brd4::Nutm1 are consistent relative to normal oral mucosa tissues among 6 samples, cMyc is only mildly upregulated and shows strong variability between samples. p63 is strongly upregulated but also to a variable degree. Sox2 is down-regulated, consistent with decreased proportions of Sox2-positive cells in mNCs compared to normal oral mucosa. No apparent correlation can be established between the expression levels of Brd4::Nutm1 and any of the three markers, consistent with the reported phenotypical variability in human NC and implies a complex regulation of NC markers for Brd4::Nutm1, meriting future investigations.*' (line 243-264)

7. Tests of statistical significance are missing from figure panels, for example Fig. 3C, Fig. 7 A,B

We added the significance testing description in the figure legends, main text and method section.

8. One of the most significant, and surprising, findings of the manuscript is the appearance of Brd4-Nutm1-driven squamous cell carcinomas in multiple tissues, including those of non-ectodermal germ layer origin. This finding warrants some more discussion. The authors make some fairly broad claims about reprogramming, for example lines 171-182 and 230-232. However, the transcriptome analysis (Fig. 3B) is superficial and highlights hallmark pathways associated with cancer in general, and is not informative about lineage-specific differentiation mechanisms. To identify the developmental processes driving squamous cell differentiation from diverse lineages in this NC mouse model, tumors could be harvested at an earlier time point, prior to the development of multiple and massive tumors. This is, in my opinion, beyond what should be required for this first manuscript. But if the authors do not undertake this type of experiment, they should temper their conclusions, for example acknowledging that Cre drivers may not be completely lineage-specific and that it is difficult to rule out any unanticipated expression of the fusion oncogene in rare cell types. Similarly, please comment on possible variable/mosaic recombination driven by NLS-Cre. Given the cellular toxicity of BRD4-NUTM1, it is surprising that mice viably develop in this model, and survive to form tumors. What accounts for the (relative) selectivity of tumor developing in this model-is there some sort of selection of which tissue types actually go on to form tumors?

Thanks for pointing this out. We changed our descriptions (lines 328-349) : *'The tumors are fast growing and appeared morphologically as poorly differentiated SCC. They presented all histopathological and molecular markers of NC (Figure 4F and Supplementary Figure 4D). As the pancreatic ductal progenitors normally initiate adenocarcinoma, and and mesenchymal progenitors normally initiate sarcomas, the ability to induce tumors morphologically consistent with poorly differentiated SCC-like tumor consistently indicated a strong reprogramming activity of BRD4::NUTM1^{49,51}. However, due to the still limited number of cases and overall late-stage tumors analyzed in this study, we cannot rule out a progressive transformation to an SCC-like phenotype at later stage of NC progression or the possibility of non-specific expression Pdx1Cre or Prrx1Cre in rare squamous epithelial progenitors.'* and line (486-488): *'Thus, it appeared that BRD4::NUTM1 has a strong reprogramming ability to drive aggressive tumors histologically consistent with a poorly differentiated SCC-like phenotype of NC regardless of lineage context.'*

Thanks for raising the point of the variable/mosaic recombination driven by the NLS-Cre. We have addressed the stochastic induction of inter-chromosome translocation when addressing Reviewer 2's first comment by showing no isolated NUTM1-positive cells in the normal oral mucosa region. We counted at least 3500 cells within the oral mucosa basal layer and didn't find a single cell expressing the fusion gene. This indicates the recombination efficiency forming the *Brd4::Nutm1* is at most one in thousands. For this reason, the whole body NLS-Cre would also only induce the translocation in a small proportion of somatic cells and would not cause general toxicity to the mouse while allowing the selection of NC-initiating cells.

9. The claim that morphologic heterogeneity explains plasticity/treatment resistance is highly speculative and not supported by direct evidence. Bromodomain inhibitors (BETi) have demonstrated efficacy in suppressing tumor growth and extending survival in both animal models and human NC trials. The NC mice could be treated with BETi to assess its potential as a preclinical model for developing new treatments (Durall et al., 2023), and to directly test the idea that different cell morphologies will be associated with selective treatment resistance.

Thanks for pointing out that the conclusion is rather speculative. We have now removed the comment about morphological heterogeneity explaining the plasticity and treatment resistance. In our opinion, functional proof of the relationship between intra- and inter-tumor heterogeneity and treatment resistance to BETi or other drugs is a long-term effort that may be outside the scope of this paper. We will address that in future projects.

Minor Comments

1. Fig. 1F: Including and analyzing low-power DC-FISH images, particularly in younger or fetal samples, could offer valuable insights into translocation frequency. The translocation frequency was addressed by our response to major comment 1. The DC-FISH is quite technically challenging, and we do not have enough statistical power to use that to calculate translocation frequency in normal tissue regions of NC-bearing mice. In the revision, DC-FISH only served as supporting evidence for the translocation, while the fusion junction was proved by more robust methods including more sanger sequencing of the junction, and junction calling from RNAseq and WGS data.

2. Fig. 1G: In panel a, the highlighted regions meant to correspond to panel b and c were erroneously labelled as "a" and "b"

Corrected.

3. Supplementary Fig. 2A: No information about the location or tissue type associated with this image was provided. Supplementary Fig. 2C: For improved clarity, consider adding a box to the low-magnification image to highlight the exact location corresponding to the zoomed-in images.

Supplementary Fig. 2A is now Figure S4B and we clarified location in main text. Supplementary Fig. 2C is now Figure S4E and we added low magnification images and squares showing the locations of the zoom-ins.

4. Line 245: The correct genotype, NLS-Cre, was mistakenly typed as "NLC-Cre" in the text. Thanks for pointing out this error, we have corrected it.

5. In NLS-Cre transgenic mice, did individual animals develop multiple tumors across different organs?

Yes, we now included the information about all tumor locations in each mice as supplementary table 1.

6. Fig. 5D: Was Alcian Blue also performed on the Pdx1Cre;NCT samples?

No, Alcian Blue was only performed on samples that appeared morphologically as chondrogenic metaplasia. And we did not find chondrogenic metaplasia in the limited number of Pdx1Cre;NCT tumors we analyzed.

7. Fig. 1C: Fig. 1C seems to be missing a description in the Results section. We added the description.

Reviewer #3 (Comments to the Authors (Required)):

Summary:

This paper by Zheng and coworkers describes the establishment of an in vivo mouse model of NUT carcinoma (NC), driven in humans mostly by the BRD4::NUTM1 gene fusion. The authors elegantly describe how the model has been engineered allowing for the induction of the chromosomal translocation resulting in the fusion. This is especially important since ectopic expression of the fusion is toxic in non-NC cell lines. By allowing for a more natural expression level due to the chromosomal translocation, a more representative model of human NUT carcinoma can be established. Using several methods, the authors show that the model can recapitulate human NUT carcinoma in mice, both in site of tumour origin and histomorphologically. This will offer a great tool for studying this disease further and developing potential novel treatments in the future. One other model has already been recently published, but the authors attempt to describe why their model may be better suited to study NC.

Main points:

Model development and validation - data is supportive

Histopathology - data is supportive

Transcriptomic investigations - data is supportive

Other comments:

Major comments:

1. Regarding the validation of the t(2;17) in mice - the authors should consider showing the

output given by Delly as a supplementary table. Was the breakpoint consistent with expectations (correct location in the introns of *Brd4* and *Nutm1*?) On what material was Sanger sequencing performed to finally validate the fusion? Are any unexpected alternative splice variants seen in RNA data? If the authors have RNA-sequenced some of the tumours, a fusion calling algorithm can be used to also check for this.

Thanks for all these very constructive comments. We now addressed as follows:

- We provided junction mapping results from both RNAseq (6 tumors) and WGS (2 tumors) analysis. Both analyses confirmed the correct junction of *Brd4::Nutm1* in NC tumors. All detected correct junctions. (Figure S2 D,E,F)
- We included Sanger sequencing results from three tumors. They all showed correct junction sequence with the remaining loxp site from the recombination. (Figure S2B)

2. In the introduction the authors state that the fusion is thought to be enough to drive human HC in an otherwise quiet genomic background. Upon WGS of the two shown tumours in supplementary Figure 1, the circos plots show anything but a quiet genome. What cutoffs were used for SVs in Delly and how were these further filtered? Have the authors considering using a second variant caller and merging the output with that of Delly? Any thoughts on why so many SNVs are detected? Is this seen in human NC as well?

Thanks for raising this very important point. Although the published reviews often state that *BRD4::NUTM1* drives NC alone in a quiet genomic landscape, there are not many genomic sequencing analyses to support that conclusion. To address this question, I looked through all case reports on PubMed about Nut carcinoma and found three studies that reported tumor mutation burden (mostly analyzed by comprehensive genomic profiling panels, with one case with Whole Genome Sequencing) of a total of 10 cases of human NUT carcinoma (Doi: 10.1186/s13023-020-01449-x; 10.1093/annonc/mdw686; 10.21037/acr-19-168). These studies reported a tumor mutation burden of >1.1 mutation/Mbs all the way to 73.81 mutations/Mbs, demonstrating a widely variable mutation burden in human NC. All three studies stated that no apparent functional mutations of tumor suppressors or well-known oncogenes were discovered. However, since there is no information on the exact mutation that was discovered in each case, we cannot determine if there are any recurrent mutations.

In our manuscript, we briefly described two pairs of WGS analyses, both from oral tumor and normal liver tissue pairs from the same mouse. The mutation burden on both tumors is 2.6 mutations/Mbs and 3.1 mutations/Mbs, respectively respectively. We did not discover any functional mutations in tumor suppressor or oncogenes or any recurrent mutations or structural variance other than the t(2;17) chromosome translocation we induced. However, this does not mean we will not detect any additional functionally important mutations that are recurrent in some subset of NCs once we analyzed more samples in the future. As for the fusions detected in WGS data, we now deposited the Delly table in GEO (GSE263567). Again, although many fusions were detected, the only recurrent one from the two tumors compared to the paired normal tissue is the *Brd4::Nutm1* fusion with the precisely designed junction sequence in our model. More fusion junctions may be detected when we or others sequence more samples. We added mutation analysis methods in the methods section.

In addition, we have now used STAR-Fusion v1.13.0 to call fusion events from the bulk RNAseq data of the 6 normal oral mucosa and tumor samples. As shown in Figure S2E and F, the transcript level calling detected multiple fusions, although less than WGS, from both normal and tumor samples. This is expected because many fusions detected in WGS might not be expressed in NC. Again, the only recurrent fusion is the *Brd4::Nutm1* with the designed junction.

Thus, the NC genome might not be as quiet as perceived, but more future work will be needed to determine whether there are cooperating mutations with *Brd4::Nutm1* to drive NC.

3. In both figure 2 and supplementary figure 2, the authors show IHC of known highly expressed proteins in NC such as MYC, TP63 and so on. Could the authors correlate this with RNA expression data for these genes? Does Sox2 show a lower expression for instance in bulk RNA if only selected regions show a high protein expression? Additionally, there is likely an error in the legend of Figure 2E - the authors refer to (b) and (c) when there are only (a) and (b) in the figure.

Thanks for pointing it out. We now included RNAseq data in Figure S4F. Indeed, Sox2 is downregulated in most tumors with one outlier in the group, consistent with its heterogeneity. MYC is mildly upregulated, this is due to that normal oral mucosa basal cells also expressed MYC. We think that MYC may not be upregulated in NC, rather, just maintained from their cell of origin, future study will be conducted to address this issue.

We have corrected the labeling error in Fig. 2E.

4. The authors show that many genes are up and downregulated in mouse NC vs. normal cells. Is there any published expression data from human NCs to correlate with? Are the same genes or pathways rather up and down regulated in human NC as well? Providing a list of these genes as a supplementary table may be helpful for future comparison studies. Additionally, the deconvoluted bulk RNA data shown in supplementary figure 3D should be further clarified. Which annotations in that set match those (even if not exactly) from figure 3C.

Thanks for making these critical points. For comparing the transcriptomics data with human NC. We think it is quite challenging now for the following reasons. In publicly accessible RNAseq data, there was only one dataset from one pleural NC from Dural et al., 2023, it's difficult to perform any statistically appropriate analysis with one sample. As NC is known to be highly variable phenotypically between human cases and across anatomical sites, it would also be difficult to compare our oral mucosa-derived tumor with this pleura cancer. All other available human data are from cell lines derived from various tumors and disease stages, and with further complications of in vitro adaptation, meaningful comparison is difficult to achieve at this point. In addition, the results of differential gene expression analysis is highly dependent on the control samples used for analysis. Our RNAseq used relatively pure oral mucosa as a control to analyze mNC derived from oral mucosa, which we consider an appropriate control. For the human primary tumor sample, normal control is lacking. For cell lines, it is hard to determine an appropriate control; many studies have used lung cancer cell lines such as A549. We felt that any conclusion at this point made by comparing our mouse NC RNAseq and human data with the stated caveats would be premature and might be misleading. We will be working on composing our well-controlled cohorts for RNAseq and hopefully will be able to answer this critical question in future work. We have deposited all raw RNAseq data on GEO GSE263558), which will be publicly accessible after the publication of our current manuscript for anyone who would like to analyze and compare the data.

For the deconvolution, we've now provided data supporting the matching of the Mus Tubare and the Cell Stem Cell dataset in Supplementary Figure S5G and clarified in figure legends.

5. The text around that part of the discussion discussing the model developed by Dural et al.

should be updated to portray that this model is not the first immunocompetent one, while still lifting the strengths of this model.

Thanks! We have edited the text:

Minor comments:

1. The authors should check for consistency in using :: to denote gene fusions as this is interchanged with the older standard of using a dash, especially in the introduction. The same applies to double checking italics style for genes and normal font for proteins and how human or mouse genes are denoted to make it clearer for the reader. An example is found in the first subtitle in the results section 3.1 - do the authors mean the human fusion or the mouse one here?

Thanks! We did a thorough check and made corrections.

2. Some sentences are poorly written or too long and not easy to understand (such as lines 55-56 in the introduction or 108-112 in the results). The authors should consider correcting and rewriting them in a clearer way.

Thanks! We've corrected them.

3. A clarification for which cells/tissues are used for each of the shown IF/IHC results in the figures would be helpful for readers who are not used to performing such stainings.

We have added these clarifications in figure legends.

4. BLI should be spelled out the first time as not all readers may be aware of what it stands for. Corrected. Thanks!

5. Genes shown to be upregulated in Figure 7A-B should be italicised. Additionally, are similar genes known to be upregulated in human NC (similarly to my previous comment)?

Changed the fonts. For human NC, similar to our response above.

6. What secondary antibodies were used for immunofluorescence? This is not clear in the methods section.

We have now included supplementary Table 2 listing all antibodies used in this study.

7. Which kit were used for cDNA library preparation for RNA sequencing?

We've added it to the method section and it is the NEBNext Ultra™ II RNA Library Prep Kit for Illumina.

8. There is an error in section 4.10 - The title described WGS but the text partially describes RNA sequencing.

Thanks for pointing out. We have corrected the description in section 4.10.

April 26, 2024

RE: Life Science Alliance Manuscript #LSA-2024-02602-TR

Dr. Bin Gu
Michigan State University
Obstetrics, Gynecology & Reproductive Science
3319 Bioengineering Building 775 Woodlot Dr.
East Lansing, MI 48824

Dear Dr. Gu,

Thank you for submitting your revised manuscript entitled "Brd4::Nutm1 fusion gene initiates NUT carcinoma in vivo". We would be happy to publish your paper in Life Science Alliance pending final revisions necessary to meet our formatting guidelines.

- please address Reviewer 3's remaining comments
- please be sure that the authorship listing and order is correct
- please add ORCID ID for the secondary corresponding author -- they should have received instructions on how to do so
- please add the Twitter handle of your host institute/organization as well as your own or/and one of the authors in our system

A. FINAL FILES:

B. MANUSCRIPT ORGANIZATION AND FORMATTING:

****It is Life Science Alliance policy that if requested, original data images must be made available to the editors. Failure to provide**

original images upon request will result in unavoidable delays in publication. Please ensure that you have access to all original data images prior to final submission.**

The license to publish form must be signed before your manuscript can be sent to production. A link to the electronic license to publish form will be available to the corresponding author only. Please take a moment to check your funder requirements.

Thank you for your attention to these final processing requirements. Please revise and format the manuscript and upload materials within 4 days.

Sincerely,

Reviewer #1 (Comments to the Authors (Required)):

The authors have addressed all my comments in a sufficient manner.

Reviewer #3 (Comments to the Authors (Required)):

This resubmission by Zheng and coworkers has addressed concerns raised by various reviewers regarding the establishment of an in vivo mouse model of NUT carcinoma (NC), driven in humans mostly by the BRD4::NUTM1 gene fusion. The authors have adequately addressed all major and minor comments I had raised. I still however have three more minor comments to raise:

1. The authors should include GEO accession numbers for uploaded data in the text at appropriate locations (for instance for the WGS data at line 689 in the methods section).
2. Once again check for proper italicisation of gene names and use of :: in gene fusions in the figure legends of figures S2 and S4.
3. There are errors in the figure legend of figure S5. Part D is mentioned twice, while the CIBERSORT data is shown in E. F is then skipped and the table at the end is labelled as G in the figure and E in the figure legend. To make the table more understandable, labelling the columns might be a good idea to show which annotations come from which reference data set. For example: Cell types (Tabula Muris) and then adding the reference Jones KB et al. 2019 above the remaining annotations. Additionally, the authors should include what type of statistical testing was carried out in the right panel of part E. Was it the same test as in the boxplot from Figure 3C?

This resubmission by Zheng and coworkers has addressed concerns raised by various reviewers regarding the establishment of an in vivo mouse model of NUT carcinoma (NC), driven in humans mostly by the BRD4::NUTM1 gene fusion. The authors have adequately addressed all major and minor comments I had raised. I still however have three more minor comments to raise:

1. The authors should include GEO accession numbers for uploaded data in the text at appropriate locations (for instance for the WGS data at line 689 in the methods section).

We have added the GEO to both the method sections about WGS and RNA-seq

2. Once again check for proper italicisation of gene names and use of :: in gene fusions in the figure legends of figures S2 and S4.

We have checked and made appropriate correction. We annotate all proteins with all capital according to this (<https://www.jci.org/kiosk/publish/genestyle>). Thus all mouse fusion protein are also BRD4::NUTM1.

3. There are errors in the figure legend of figure S5. Part D is mentioned twice, while the CIBERSORT data is shown in E. F is then skipped and the table at the end is labelled as G in the figure and E in the figure legend. To make the table more understandable, labelling the columns might be a good idea to show which annotations come from which reference data set. For example: Cell types (Tabula Muris) and then adding the reference Jones KB et al. 2019 above the remaining annotations. Additionally, the authors should include what type of statistical testing was carried out in the right panel of part E. Was it the same test as in the boxplot from Figure 3C?

Thanks for pointing out. We made corrections to all these and changed accordingly. We provided the statistics description for Figure S5E right part in the legend.

April 29, 2024

RE: Life Science Alliance Manuscript #LSA-2024-02602-TRR

Dr. Bin Gu
Michigan State University
Obstetrics, Gynecology & Reproductive Science
3319 Bioengineering Building 775 Woodlot Dr.
East Lansing, MI 48824

Dear Dr. Gu,

Thank you for submitting your Research Article entitled "Brd4::Nutm1 fusion gene initiates NUT carcinoma in vivo". It is a pleasure to let you know that your manuscript is now accepted for publication in Life Science Alliance. Congratulations on this interesting work.

DISTRIBUTION OF MATERIALS:

Again, congratulations on a very nice paper. I hope you found the review process to be constructive and are pleased with how the manuscript was handled editorially. We look forward to future exciting submissions from your lab.

Sincerely,
